# Optimal transport based adversarial patch to leverage large scale attack transferability

**Pol Labarbarie**[1,2] **Adrien Chan-Hon-Tong**[2] **Stéphane Herbin**[2] **Milad Leyli-Abadi**[1]
[1]IRT-SystemX, [2]ONERA/DTIS, University Paris-Saclay
Palaiseau, France
`{firstname.name}@irt-systemx.fr`
`{firstname.name}@onera.fr`

## ABSTRACT

Adversarial patch attacks, where a small patch is placed in the scene to fool neural networks, have been studied for numerous applications. Focusing on image classification, we consider the setting of a black-box transfer attack where an attacker does not know the target model. Instead of forcing corrupted image representations to cross the nearest decision boundaries or converge to a particular point, we propose a distribution-oriented approach. We rely on optimal transport to push the feature distribution of attacked images towards an already modeled distribution. We show that this new distribution-oriented approach leads to better transferable patches. Through digital experiments conducted on ImageNet-1K, we provide evidence that our new patches are the only ones that can simultaneously influence multiple Transformer models and Convolutional Neural Networks. Physical world experiments demonstrate that our patch can affect systems in deployment without explicit knowledge.

## 1 INTRODUCTION

Deep neural networks have shown vulnerability to adversarial examples, *i.e.*, norm-bounded perturbations of their inputs designed to fool them (Szegedy et al., 2013; Biggio, 2013). Such vulnerability has motivated researchers to develop empirical robustification methods (Madry et al., 2017) or to provide some theoretical robustness guarantees (Cohen, 2019). Other research is dedicated to designing more powerful attacks (Kurakin et al., 2016). These attacks are invisible patterns added to the whole image – a pixel array – which, therefore, has to be accessible: a strong practical limitation.

Adversarial *patch* attacks (APA) are a more realistic type of attack expected to be realizable in the physical world. They rely on adding a small textured patch to the scene. Since such a patch can be easily printed and localized on an object or in the environment, it poses a serious threat in various contexts and related tasks. For example, Brown et al. (2017) produce a patch capable of fooling multiple ImageNet-1K classification models. Patch attacks can also threaten other visual tasks (Thys et al., 2019; Lee & Kolter, 2019; Saha et al., 2020; Hu et al., 2022; Nesti et al., 2022). Saha et al. (2020) design a patch which, when placed on a stop sign or the roadway, may result in the missed detection of a pedestrian crossing the road. Another APA for a semantic segmentation task is proposed in (Nesti et al., 2022), which reduces the baseline model accuracy. Despite the good attacking performance of current APA in whitebox configuration (applied on the same model that they have been learned), their effectiveness is mitigated when transferring to unseen models (blackbox configuration). Prior works focus either on studying the whitebox performance of their patch against Transformer architecture (Fu et al., 2022; Lovisotto et al., 2022) or on studying the whitebox and/or the blackbox performance of their patch against classical CNNs (Brown et al., 2017; Karmon et al., 2018; Liu et al., 2019; Doan et al., 2022; Casper et al., 2022). Focusing on image classification, we propose a new attack perspective to improve the transfer of patch attacks to unseen models.

Most previous work on APA for classification influence the network to output a target class with high confidence (Brown et al., 2017; Karmon et al., 2018; Liu et al., 2019; Doan et al., 2022; Casper et al., 2022). This strategy involves pushing the deep representation of images to cross the nearest decision boundary of the source model. The strategy has two drawbacks: it is highly dependent on

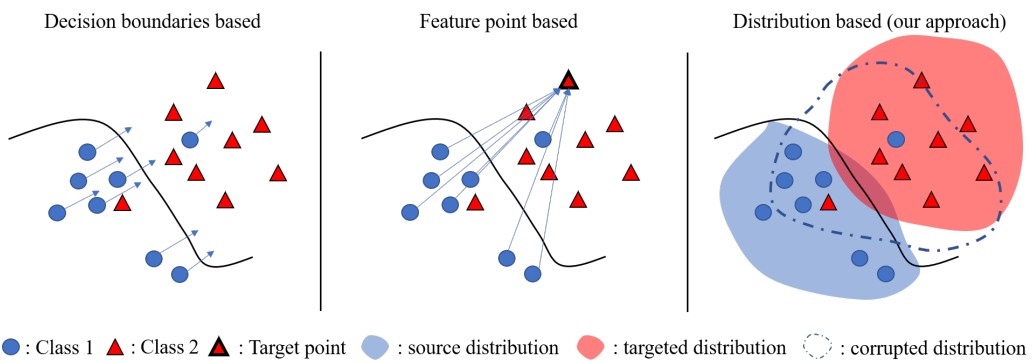

Figure 1: Three different strategies for designing patch attacks. Left: The attack pushes multiple samples to the other side of a decision boundary defined for a particular model. Middle: The attack matches a given point in the feature space that is expected to represent a sample from a different class. Right: Our strategy narrows the distribution gap between the samples corrupted by the patch and another misleading distribution in feature space. It does not depend on the decision boundaries, nor on the choice of a specific target point in the feature space.

the model on which the attack is based, and the patch may push the corrupted image representations into unknown regions of the representation space. Instead of blindly maximizing the probability of a target class, research on invisible adversarial examples suggests considering the feature space of deep networks (Zhou et al., 2018; Rozsa et al., 2017; Inkawhich et al., 2019; 2020). For example, Inkawhich et al. (2019) propose optimizing the adversarial examples to match the deep feature representations of an existing target image.

To overcome the dependence of the attack on a single decision boundary of the source model and to relax the specificity of the selection of the target feature point, we propose a distribution-oriented approach. The learning principle of our patch attack is to globally alter the feature distribution of a set of images from a particular class to match another known distribution to be taken from another class. To do so, we propose to optimize with respect to the Wasserstein loss which has several practical and theoretical advantages (Peyré et al., 2019; Frogner et al., 2015; Arjovsky et al., 2017). The role of the patch, when placed in the scene, is to push the feature distribution towards this known misleading class distribution (Fig 1). Such a global strategy in feature space is expected to allow a better transferability capability, as it is independent of the decision boundary constructed by the classifier and the choice of the target point (as proposed by Inkawhich et al. (2019)). By conducting extensive experiments on ImageNet-1K (Deng et al., 2009), we indeed show that our APA is more model transferable and is more physically feasible than previous APAs for a large ensemble of network architectures, including classical CNNs (Simonyan & Zisserman, 2014; Szegedy et al., 2016; He et al., 2016; Huang et al., 2017), recent CNNs (Tan & Le, 2019; Liu et al., 2022) and Vision Transformers (Touvron et al., 2021; Liu et al., 2021).

The main contributions of this work are to:

- introduce a new framework based on optimal transport for creating patch attacks that are highly transferable to unknown networks. This framework is based on the idea of attacking feature distributions, which is independent to the classifier decision boundaries and more robust to optimization artifacts than the feature point method;

- show that our attack works for the most extensive spectrum of deep networks considered in the patch attack literature: we deal with various versions of Convolutional Neural Networks, Transformers, and adversarially trained models and show transferability superiority through extensive experiments on ImageNet-1K;

- provide digital and physical experiments demonstrating that our patch is potentially harmful in the physical world (including against state of the art defense).

## 2 RELATED WORK

### 2.1 TRANSFERABLE INVISIBLE ADVERSARIAL EXAMPLES

After discovering adversarial examples, some works studied whether these invisible adversarial attacks were transferable from one model to another. To enhance the adversarial examples transferability, Inkawhich et al. (2019) consider the feature space of deep networks rather than their decision space. The rationale for such a strategy is that the feature space is expected to capture the useful information in an image more universally, allowing better attack transferability between models. They propose to build the attack to push the corrupted image features close to a single specific feature point (feature representation of a chosen sample of a pre-chosen class). To strengthen the transferability Inkawhich et al. (2020) train offline multiple specific auxiliary classifiers. Rather than independently train each adversarial examples by iterative methods, Poursaeed et al. (2018); Naseer et al. (2019a; 2021); Zhao et al. (2023) train a Generative Adversarial Network (GAN) to transform clean images to adversarial examples. Naseer et al. (2021) (called TTP method) learn their GAN by minimizing the Kullback-Leibler divergence between the probability-class distribution of adversarial examples and the probability-class distribution of target class images. From a generalization error bound for black-box targeted attacks, Zhao et al. (2023) derive an algorithm to train their GAN to minimize the maximum model discrepancy (M3D method) between two models. However, all these previously presented works study only the transferability of invisible adversarial examples, and, it is not clear that they work for APA.

### 2.2 ADVERSARIAL PATCH ATTACK

APAs were first introduced for image classification by Brown et al. (2017). Rather than finding a small additive adversarial noise, they instead constrain the optimization procedure to a small part of the image while also allowing the optimization to be unconstrained in magnitude. They design a patch (called GAP) by maximizing, under patch transformations, the log Softmax of a selected target class.

The resulting patch was capable of fooling five ImageNet-1K classification models. To increase the fooling effectiveness of the patch, Karmon et al. (2018) (LaVAN method) add a new term to the loss criterion initially proposed by Brown et al. (2017). By minimizing the log Softmax of image ground truth, this new term ensures the misclassification of the attacked image. Other methods propose to use generative-based approaches: Liu et al. (2019) (PS-GAN method) train a GAN to generate a background-harmonious patch that enhances both the visual fidelity and attacking ability of the patch. Instead of training a GAN, Doan et al. (2022) and Casper et al. (2022) use a

Table 1: Typology of Adversarial Patch Attacks (APA) papers and transferable invisible adversarial examples works depending on the requirement to illustrate a transferable physical APA. Each column represents a essential characteristic to demonstrate the real-world criticality of an APA. The symbols ✓, ≈ and ∅ represent "measured", "ambiguous" and "not evaluated", respectively.

| | Transfer | | Defended networks | APA |
|---|---|---|---|---|
| | Narrow | Broad | | |
| GAP (Brown et al., 2017) | ✓ | ∅ | ∅ | ✓ |
| LaVAN (Karmon et al., 2018) | ∅ | ∅ | ∅ | ✓ |
| PS-GAN (Liu et al., 2019) | ≈ | ∅ | ∅ | ✓ |
| TnT (Doan et al., 2022) | ✓ | ∅ | ✓ | ✓ |
| Casper et al. (2022) | ≈ | ∅ | ≈ | ✓ |
| Inkawhich et al. (2019) | ✓ | ∅ | ✓ | ∅ |
| TTP Naseer et al. (2021) | ✓ | ∅ | ✓ | ∅ |
| M3D Zhao et al. (2023) | ✓ | ∅ | ✓ | ∅ |
| **Ours** | ✓ | ✓ | ✓ | ✓ |

pre-trained GAN directly. To change the generated flower to an adversarial flower, Doan et al. (2022) (TnT method) modify the latent representation of the generator. Casper et al. (2022) perturb the latent representation at some chosen generator layer. All the above works optimize their patch to maximize the log Softmax of some classifiers.

### 2.3 TRANSFERABILITY EVALUATION

The previously presented APAs Brown et al. (2017); Karmon et al. (2018); Liu et al. (2019); Doan et al. (2022); Casper et al. (2022) use different evaluation settings in their experimentation. Some of these works measure the transferability of their patch to unknown models. Brown et al. (2017) and Doan et al. (2022) measure their patch transferability but only consider dated CNNs like ResNet trained with a less generalizing learning policy than recent ones. The GAP patch is also applied at random locations in these images, giving rise to possible occlusion of the main object. The PS-GAN

generated patch (Liu et al., 2019) shows transferability among unseen models. However, the evaluated models are not state-of-the-art, and the experiments are conducted on the small GTRSB dataset (Stallkamp et al., 2011). Focused on the interpretability of decision-making of deep networks, Casper et al. (2022) do not provide quantitative results on network transferability. Table 1 contextualizes our approach with respect to others.

Our work bridges the gap between the transferability studies of invisible adversarial examples and adversarial patch attacks. We follow the principle that the attack should be designed with an optimization metric defined in the feature space rather than in the decision space (Inkawhich et al., 2019). Moreover, we generalize the point-wise strategy proposed by Inkawhich et al. (2019), which presents several drawbacks. First, optimizing when the objective is to push multiple points to a unique point is likely to fail. When the optimization succeeds, the power of the attack depends highly on the choice of the target point (see Appendix B). Furthermore, a single feature point can be well-classified by one network but misclassified by another, thus limiting the transferability of the attack to multiple networks. We propose to avoid target point selection and to broaden the range of features under attack by considering target point distributions instead. This is done through optimal transport for the loss to be optimized.

## 3 METHODOLOGY

### 3.1 BACKGROUND

The theory of optimal transport (Peyré et al., 2019; Villani et al., 2009) provides several techniques for efficient computation of distances between distributions. It has been shown that optimizing with respect to the Wasserstein loss has various practical benefits over the KL-divergence loss (Peyré et al., 2012; Frogner et al., 2015; Arjovsky et al., 2017; Gulrajani et al., 2017). Unlike the KL-divergence and its related dissimilarity measures (e.g. Jensen-Shannon divergence), the Wasserstein distance can provide a meaningful notion of closeness (i.e. distance) for distributions supported on non-overlapping low dimensional manifolds.

Let $\mathcal{P}_p(\mathbb{R}^d) = \{\mu \in \mathcal{P}(\mathbb{R}^d) : \int_{\mathbb{R}} ||x||^p \mathrm{d}\mu(x) < \infty\}$ be the set of probability measures on $\mathbb{R}^d$ with finite moment of order $p$, with $p \in [1, +\infty)$. The $p$-Wasserstein distance is defined as

$$\mathbf{W}_p^p(\mu, \nu) = \inf_{\pi \in \Pi(\mu,\nu)} \int_{\mathbb{R}^d \times \mathbb{R}^d} ||x - y||^p \mathrm{d}\pi(x, y), \tag{1}$$

where $\mu, \nu \in \mathcal{P}_p(\mathbb{R}^d)$, $||.||$ is the Euclidean norm and $\Pi(\mu, \nu)$ is the set of probability measures on $\mathbb{R}^d \times \mathbb{R}^d$ whose marginals with respect to the first and second variables are given by $\mu$ and $\nu$ respectively. The quantity $\mathbf{W}_p(\mu, \nu)$ in not analytically avaible in general. To solve Eq 1, the standard methods are linear programs and have a worst-case computational complexity in $\mathcal{O}(n^3 \log(n))$ where $n$ is the number of samples (Peyré et al., 2019).

To leverage the computational efficiency of Eq 1 ,Rabin et al. (2012); Bonneel et al. (2015) define a new metric named Sliced-Wasserstein distance. This new metric is based on the fact that for one-dimensional probability measure the $p$-Wasserstein distance (1) has the following closed-form

$$\mathbf{W}_p^p(\mu, \nu) = \int_0^1 |Q_\mu(s) - Q_\nu(s)|^p \mathrm{d}s, \tag{2}$$

where $Q_\mu$ and $Q_\nu$ are the quantile functions of $\mu$ and $\nu$ respectively. Let $\mathbb{S}^{d-1}$ be the $d$-dimensional unit sphere and $\sigma$ the uniform distribution on $\mathbb{S}^{d-1}$. For $\theta \in \mathbb{S}^{d-1}$, we define the linear form for all $x \in \mathbb{R}^d$ by $\theta^*(x) = \langle \theta, x \rangle$. The Sliced-Wasserstein distance is then defined by

$$\mathbf{SW}_p^p(\mu, \nu) = \int_{\mathbb{S}^{d-1}} \mathbf{W}_p^p(\theta_\sharp^*\mu, \theta_\sharp^*\nu) \mathrm{d}\sigma(\theta), \tag{3}$$

where $\mu, \nu \in \mathcal{P}_p(\mathbb{R}^d)$, $p \in [1, +\infty)$ and $\theta_\sharp^*\mu$ and $\theta_\sharp^*\nu$ are the push-forward by $\theta^*$ of $\mu$ and $\nu$ respectively. In practise, Eq 3 is approximated with a standard Monte Carlo method. We denote by $\mathbf{SW}_p^p(\mu, \nu)_K$ its numerical approximation where $K$ is the number of random projections. Since $\theta_\sharp^*\mu$ and $\theta_\sharp^*\nu$ are univariate distributions, the resulting complexity of the approximation is general more efficient than resolving Eq 1. The corresponding computational complexity is $\mathcal{O}(Kdn + Kn \log(n))$. We show also in section A of Appendix that the empirical computation time of our approach remains similar to other methods.

## 3.2 OPTIMAL TRANSPORT BASED LOSS

We consider the standard notation where $(x_i, y_i) \in \mathcal{X} \times \mathcal{Y}$, $i = 1, ..., n$, are samples drawn from a joint distribution of random variables $X$ and $Y$. We are considering an image classification problem, where the input is sampled from $\mathcal{X} = \mathbb{R}^{h \times w \times c}$, where $h \times w$ are the image dimensions and $c$ is the number of channels and where the output $\mathcal{Y}$ is sampled from a set of M labels $\{y_1, ..., y_M\}$. Let $F : \mathcal{X} \to \mathcal{Y}$ be a given pre-trained neural network that we wish to attack. We assume that the functional architecture of $F$ follows a classical encoder-decoder schema. We denote by $f$ the encoder part of $F$, where $f$ is composed by a set of $J$ layers; $\mathcal{L} = \{l_1, ..., l_J\}$. Except for the last layer which usually directly result from an average pooling layer, we apply an average pooling layer to obtain a feature vector. For all $l \in \mathcal{L}$, $f^{(l)}$ maps $x \in \mathcal{X}$ to the feature space $\mathcal{S}^{(l)} = \mathbb{R}^{c_l}$, where $c_l$ is the number of channels.

For a given target class $y$, we denote by $\nu_y^{(l)}$ the multivariate target distribution of $f^{(l)}(X)$ when the class of $X$ is $y$. The principle of our proposed method is to design a patch by moving the corrupted image distribution towards the target $\nu_y^{(l)}$ and solve:

$$\delta^* = \arg\min_\delta \mathbb{E}_X \left[ \sum_{l \in \mathcal{L}} OT(\mu_{X_\delta}^{(l)}, \nu_y^{(l)}) \right], \tag{4}$$

where $\mu_{X_\delta}$ is the estimated feature distribution of the corrupted source images and $OT$ could be $\mathbf{W}_p^p$ or $\mathbf{SW}_p^p$.

In practice, we solve a regularized version of Eq 4 using Expectation over Transformations (EoT) from Athalye et al. (2018). This regularization makes patches more physically realizable. Let $\mathcal{T}$ be a distribution over transformation (*e.g.*, rotations, scaling, blur, ...) and $E$ a distribution over locations. Following Brown et al. (2017); Casper et al. (2022) we denote by $A(\delta, x, e, t)$ the patch applicator operator in an image $x$ where $\delta$ is the patch, $t$ are patch transformations and $e$ is the patch location in the image $x$. Our patch is therefore trained to optimize the following objective:

$$\delta^* = \arg\min_\delta \mathbb{E}_{X, t \sim \mathcal{T}, e \sim E} \left[ \sum_{l \in \mathcal{L}} OT(\mu_{A(\delta, X, e, t)}^{(l)}, \nu_y^{(l)}) + TV(\delta) \right], \tag{5}$$

where $TV$ is the total variation loss discouraging high-frequency patterns. We will denote by $(\mathbf{W}_p^p)^{(N)}$ and $(\mathbf{SW}_p^p)^{(N)}$ when we attack $N$ layers by solving the standard or the sliced version of the Wasserstein distance respectively. We choose by convention that for $N = 1$, $l = l_J$, *i.e.*, we are attacking the last layer of $f$.

## 4 EXPERIMENTS

This section evaluates our APA through digital, hybrid and physical world experiments. In all experiments, the objective is to craft an APA with a high targeted success rate (tSuc).

We consider the single-source model setting and test attacking transferability to other models. Transferability is tested between ImageNet-1K (Deng et al., 2009) models; ResNet 18/34/50-V1/50-V2 (He et al., 2016), DenseNet 121/161/169/201 (Huang et al., 2017), EfficientNet B0/B1/B2/B3/B4 (Tan & Le, 2019), ConvNext Tiny/Small (Liu et al., 2022), VGG19 (Deng et al., 2009), Inception-V3 (Szegedy et al., 2016) and Swin Tiny/Small/Base (T/S/B) (Liu et al., 2021) from Pytorch Model Zoo, DeiT T/S/B from Timm Model Zoo, ResNet50 ReLU Adv and DeiT S Adv, adversarially trained models (trained against invisible adversarial examples) from Bai et al. (2021) and finally ResNet50 self-supervised learned from Caron et al. (2020). We regroup these models into the following categories depending on their architecture and their training recipes: CNNs-V1 = {ResNet 18/34/50-V1, DenseNet 121/161/169/201, VGG19, Inception-V3}, CNNs-V2 = {ResNet 50-V2/50-self}, ENet = {EfficientNet B0/B1/B2/B3/B4}, CNext = {ConvNext T/S}, DeiT = {DeiT T/S/B}, Swin = {Swin T/S/B} and AT = {ResNet50 ReLU Adv and DeiT S Adv}, where AT stands for Adversarially trained.

**Evaluated methods.** We consider GAP (Brown et al., 2017), LaVAN (Karmon et al., 2018), TnT (Doan et al., 2022), Casper et al. (2022) and Logit (Zhao et al., 2021) as decision boundary-based

baselines. Because of its ease of computation compared to Inkawhich et al. (2020), which requires off-line training of multiple specific auxiliary models, we choose to adapt the proposed method by Inkawhich et al. (2019) as a baseline (we name it L2) to craft an APA based on attacking the feature space. We also adapt the recent state-of-the-art works on transferable invisible adversarial examples (Naseer et al., 2021; Zhao et al., 2023) to craft an APA. To do so, we convert generative methods (Naseer et al., 2021; Zhao et al., 2023) to iterative ones: the objective is to create one universal APA and not one for each image.

**Experimental setup.** For the sake of comparison, baselines and our method are crafted using the same training recipes. To control the balance between the adversarial loss and the total variation loss, the gradient of each loss is computed individually, normalized, and combined using a weighted sum. Patch values are clipped into the image range at each iteration. Following prior works Brown et al. (2017); Lee & Kolter (2019); Casper et al. (2022), we choose and fix the sampling distributions from EoT (Eykholt et al., 2018) for all the methods. During training and evaluation, patches are randomly placed to the side of images (to avoid occluding the object of interest), and transformations and noises are applied to the patch to mimic real-world situations. Appendix E evaluates the robustness of models according to the patch position in the image. We randomly choose nine targeted classes (see Appendix A for details) and design a patch to fool the network targeting each of these classes. We split the ImageNet-1K validation set into a training set of 40000 images on which we train patches and a test set of 10000 images on which we evaluate their impact. The patch optimization is performed using 100 epochs (1 epoch equals 1000 iterations) with a batch size of 50 images and for three different learning rates (0.1, 0.5, 1). We choose for our method $p = 2$ and $K = 500$ (reasons are explained in Appendix I). We evaluate the APA with the best loss among the three learning rates, leading to one patch per method and per class. Finally, reported tSuc are the average over the classes and patch sizes (from $70 \times 70$ to $90 \times 90$ which is the standard setting consider in the literature (Brown et al., 2017; Poursaeed et al., 2018; Doan et al., 2022; Casper et al., 2022)).

## 4.1 DIGITAL EXPERIMENTS

### 4.1.1 TRANSFERABILITY AMONG NETWORKS

We select from the previously defined families the following models: ResNet34, ResNet50-V1, ResNet50-V2, ResNet50-self, EfficientNet-B0, ConvNext-S, DeiT-S, Swin-T, Swin-S, Swin-B. We design patches to attack one of these source models. Then we measure the attacking transferability when the resulting patch is used to fool the remaining models (target models). For example, patches trained and tested on the Swin family provide three patches. Each one is trained individually on Swin-T, Swin-S or Swin-B, and is evaluated against the two other Swin models. Table 2 summarizes, for each method, the best transferring attack performance. We select and report the results of the source family producing the highest mean targeted success rate (tSuc, rate at which the attacked images are classified as the patch target label) according to the method. Our method shows the best transferability capacity: highest mean, min and max tSuc.

Table 2: Best transfer results from a single model to all others obtained for each method (tSuc (%) higher is better for an attack).

| Method | min | mean | max |
|---|---|---|---|
| GAP (Brown et al., 2017) | 2.22 | 15.46 | 37.33 |
| LaVAN (Karmon et al., 2018) | 2.26 | 8.67 | 31.4 |
| L2 (Inkawhich et al., 2019) | 4.44 | 13.6 | 32.78 |
| TnT (Doan et al., 2022) | 0.67 | 2.11 | 5.84 |
| Casper et al. (2022) | 0.33 | 3.81 | 14.85 |
| Logit Zhao et al. (2021) | 2.22 | 7.55 | 26.55 |
| TTP Naseer et al. (2021) | 2.33 | 13.77 | 31.87 |
| M3D Zhao et al. (2023) | 0.84 | 5.19 | 17.11 |
| **Ours $(\mathbf{SW}_2^2)_{500}^{(1)}$** | **8.93** | **22.56** | **45.31** |
| **Ours $(\mathbf{W}_2^2)^{(1)}$** | **8.09** | **21.14** | **49.1** |

Table 3 details transferability results for all source families. From this table, we can make several conclusions: Networks trained with older training recipes (CNNs-v1) seem more vulnerable to attacks regardless of the attacking procedures. These networks are more sensitive to salient patches present in the image. As presented in Bai et al. (2021), new training recipes (scheduler, augmenting training data like RandAug and Mixup, ...) appear to robustify models for convolutional networks and transformers. Baseline methods were not able to create a patch that can strongly transfer to CNext and Swin models even when these patches are learned using the same model category. This indicates that rather than catch the useful common

information in deep networks, baseline methods produce a patch that tends to overfit on the specific weights of the model. This is particularly the case for decision boundaries-based APAs (GAP (Brown et al., 2017), LaVAN (Karmon et al., 2018), TnT (Doan et al., 2022) and Casper et al. (2022)). Naseer et al. (2021) method seems to create a patch that better captures the overall source model decision boundaries. This method leads to better results than decision boundaries-based APAs on the DeiT family of networks (networks with more complex representations than CNNs-V1 networks). Creating a patch that minimizes the maximum discrepancy between two models (Zhao et al., 2023) is unstable and generally results in a patch that is not transferable. A possible explanation is that APA induces a higher shift in the feature space than invisible adversarial examples.

A patch resulting from an optimization defined in the feature space reduces the patch overfitting and increases the transferability to other networks. This suggests that the patch has learned more about the common information to model the different classes rather than trying to cross the decision boundaries. However, the L2 methodology (Inkawhich et al., 2019) is unstable and is highly dependent on the choice of the target point, resulting in lower performance than our method (see Appendix B). Our two methods (exact and sliced version) outperform the previous methods on transferability. We remark that patches learned using Swin or CNext seem more universal as they can transfer to multiple models. When crafted on Swin models, we produce a patch capable of transferring uniformly well to all the models. We show in Appendix F that an ensemble of CNNs-v1 models can not reach the level of transferability obtained by our method when targeting Swin models. These results indicate that our method allows the patch to learn more about the common information shared across networks. The following experiments are performed on Swin patches as they lead to a more uniform transferability across networks.

### 4.1.2 EFFECTIVENESS AGAINST ROBUSTIFIED NETWORKS

We now consider a more realistic scenario in which the attacked system uses a defense mechanism. We propose to use Local Gradients Smoothing (LGS) (Naseer et al., 2019b), as it is one of the strongest defense mechanism against patch attacks. LGS smooths salient regions in images before feeding them to the network. We reproduce the previous experiments for three different smoothing factors $\lambda \in \{1.5, 1.9, 2.3\}$ for LGS while fixing other parameters as in the article (we report here results for $\lambda = 1.5$, see K for other results). For each method, we evaluate their Swin patches against networks robustified by LGS. Our method achieves the best transfer results demonstrating the criticality of our attack even for robustified networks (Table 4). We now suppose the target network has been adversarially trained (AT) against invisible adversarial examples. The patch at-

Table 5: Transfer results of digital, scanned and scanned defended patches (mean tSuc (%)). Patches are designed on Swin models and for class bird house.

|  | Digital | Scan | Scan Defended |
| --- | --- | --- | --- |
| Clean | 0.1 | 0.1 | 0.1 |
| GAP Brown et al. (2017) | 1.3 | 1.1 | 0.92 |
| LaVAN Karmon et al. (2018) | 1.53 | 1.05 | 0.88 |
| L2 Inkawhich et al. (2019) | 8.65 | 4.54 | 4.26 |
| TnT (Doan et al., 2022) | 0.78 | 0.43 | 0.37 |
| Casper et al. (2022) | 1.13 | 0.49 | 0.39 |
| TTP (Naseer et al., 2021) | 1.06 | 0.73 | 0.52 |
| M3D (Zhao et al., 2023) | 1.84 | 1.08 | 0.68 |
| **Ours($\mathbf{SW}_2^2)_{500}^{(1)}$** | **19** | **12.41** | **12.11** |
| **Ours($\mathbf{W}_2^2)^{(1)}$** | **20.04** | **12.59** | **12.41** |

tacks which are not learned on AT models could reduce their accuracy when transferred to these models. However, the AT models do not get fooled by the patch to predict the targeted class. (clean accuracy: 65.44, attacked accuracy: 57.65, tsuc: 0.72). AT models seem to have different class representations and are hard to force to predict a chosen class. When designed on one AT model and transferred to another model, our patches and GAP patches produce the best transfer performances (see Appendix G).

### 4.2 HYBRID EXPERIMENTS

In this section, we propose to measure the physicality of patches through a hybrid experiment and to simulate the potential effect of patches in the real world. We consider the following steps: printing and digitalization. Scanned patches are placed numerically in images using the same procedure as in the previous section (physical transformations are applied to them). We use patches designed on Swin-T and the results for three different settings (i.e., digital, scan and scan with defense) are

Table 3: Transfer results (tSuc (%), higher is better attack) between categories of models. Results are averaged over classes, over patch sizes and over networks within a category. Patches are placed randomly in the image without object overlapping. Physical transformations (*e.g.*, noise, rotations) are applied to patches. Control stands for inserting a real object of the corresponding class as a patch.

| Method | Source | CNNs-v1 | CNNs-v2 | ENet | CNext | DeiT | Swin | mean / std |
|---|---|---|---|---|---|---|---|---|
| | Clean | 0.1 | 0.1 | 0.1 | 0.1 | 0.1 | 0.1 | 0.1 / 0 |
| Control | | 2.85 | 1.59 | 0.86 | 0.54 | 1.57 | 0.93 | 1.39 / 0.75 |
| GAP (Brown et al., 2017) | CNNs-v1 | 36.61 | 9.64 | 5.54 | 2.43 | 4.03 | 3.05 | 10.22 / 12.04 |
| | CNNs-v2 | 15.6 | 9.57 | 3.66 | 2.74 | 3.51 | 2.49 | 6.26 / 4.82 |
| | ENet | 37.33 | 10.11 | 29.88 | 2.22 | 8.91 | 4.3 | 15.46 / 13.28 |
| | CNext | 0.33 | 0.77 | 0.23 | 0.97 | 0.43 | 0.8 | 0.59 / 0.27 |
| | DeiT | 1.43 | 1.97 | 0.46 | 1.25 | 11.54 | 3.58 | 3.37 / 3.78 |
| | Swin | 1.46 | 1.54 | 0.66 | 1.33 | 1.58 | 6.15 | 2.12 / 1.83 |
| LaVAN (Karmon et al., 2018) | CNNs-v1 | 31.4 | 8.56 | 4.32 | 2.26 | 2.49 | 3.01 | 8.67 / 10.38 |
| | CNNs-v2 | 11.08 | 9.68 | 2.33 | 2.45 | 2.24 | 2.13 | 4.98 / 3.84 |
| | ENet | 8.74 | 4.76 | 11.31 | 1.08 | 3.33 | 2.58 | 5.3 / 3.59 |
| | CNext | 0.45 | 0.63 | 0.26 | 0.44 | 0.47 | 0.87 | 0.52 / 0.19 |
| | DeiT | 2.1 | 1.53 | 0.93 | 0.61 | 5.84 | 2.45 | 2.24 / 1.73 |
| | Swin | 1.45 | 1.41 | 0.57 | 1.31 | 1.29 | 9.44 | 2.58 / 3.08 |
| L2 (Inkawhich et al., 2019) | CNNs-v1 | 14.76 | 5.06 | 2.69 | 0.88 | 1.78 | 1.08 | 4.37 / 4.85 |
| | CNNs-v2 | 3.25 | 3.5 | 0.64 | 1.44 | 0.57 | 1.04 | 1.74 / 1.19 |
| | ENet | 14.33 | 4.12 | 13.35 | 0.79 | 3.02 | 1.88 | 6.25 / 5.47 |
| | CNext | 2.46 | 9.66 | 0.92 | 20.2 | 1.73 | 10.67 | 7.6 / 6.81 |
| | DeiT | 17.88 | 10.23 | 8.15 | 4.44 | 32.78 | 8.1 | 13.6 / 9.5 |
| | Swin | 8.2 | 8.54 | 3.22 | 7.24 | 5.38 | 23.24 | 9.3 / 6.49 |
| TnT (Doan et al., 2022) | CNNs-v1 | 5.84 | 1.5 | 2.12 | 0.67 | 1.43 | 1.08 | 2.11 / 1.73 |
| | CNNs-v2 | 1.82 | 0.69 | 0.59 | 0.37 | 0.52 | 0.6 | 0.77 / 0.48 |
| | ENet | 2.13 | 0.92 | 1.4 | 0.43 | 0.71 | 0.64 | 1.04 / 0.57 |
| | CNext | 0.48 | 0.49 | 0.24 | 0.32 | 0.3 | 0.4 | 0.37 / 0.09 |
| | DeiT | 1.12 | 0.85 | 0.61 | 0.58 | 2.43 | 1.03 | 1.1 / 0.63 |
| | Swin | 1.41 | 1.03 | 0.55 | 0.81 | 1.61 | 1.68 | 1.18 / 0.42 |
| Casper et al. (2022) | CNNs-v1 | 12.87 | 1.62 | 1.2 | 0.28 | 0.33 | 0.19 | 2.75 / 4.56 |
| | CNNs-v2 | 7.8 | 7.44 | 1.26 | 0.83 | 0.95 | 0.78 | 3.17 / 3.15 |
| | ENet | 5.37 | 0.85 | 14.85 | 0.33 | 0.68 | 0.78 | 3.81 / 5.23 |
| | CNext | 0.42 | 0.28 | 0.22 | 0.45 | 0.15 | 0.22 | 0.29 / 0.11 |
| | DeiT | 2.86 | 1.38 | 0.98 | 0.91 | 10.19 | 2.22 | 3.09 / 3.25 |
| | Swin | 0.56 | 0.4 | 0.35 | 0.52 | 0.32 | 1.87 | 0.67 / 0.54 |
| TTP (Naseer et al., 2021) | CNNs-v1 | 35.4 | 8.41 | 5.43 | 1.58 | 3.46 | 2.29 | 9.43 / 11.83 |
| | CNNs-v2 | 17.55 | 9.67 | 3.3 | 3.87 | 3.66 | 3.87 | 6.99 / 5.21 |
| | ENet | 31.87 | 8.88 | 27.16 | 2.33 | 8.75 | 3.65 | 13.77 / 11.47 |
| | CNext | 0.49 | 3.39 | 0.22 | 7.87 | 0.48 | 2.71 | 2.53 / 2.67 |
| | DeiT | 3.87 | 3.24 | 1.51 | 1.64 | 13.75 | 3.85 | 4.64 / 4.18 |
| | Swin | 1.53 | 1.22 | 0.53 | 0.98 | 1.18 | 5.54 | 1.83 / 1.69 |
| Zhao et al. (2023) | CNNs-v1 | 17.11 | 6.18 | 3.59 | 0.84 | 1.98 | 1.43 | 5.19 / 5.61 |
| | CNNs-v2 | 7.45 | 11.77 | 1.77 | 2.13 | 1.51 | 2.33 | 4.49 / 3.84 |
| | ENet | 11.21 | 1.97 | 3.34 | 0.54 | 1.0 | 0.88 | 3.16 / 3.72 |
| | CNext | 0.34 | 0.36 | 0.16 | 0.21 | 1.78 | 0.28 | 0.52 / 0.57 |
| | DeiT | 2.39 | 1.59 | 0.81 | 1.07 | 7.7 | 2.9 | 2.74 / 2.33 |
| | Swin | 1.85 | 1.63 | 0.71 | 0.82 | 1.55 | 3.6 | 1.69 / 0.95 |
| **Ours ($\mathbf{SW}_2^2)_{500}^{(1)}$** | CNNs-v1 | 25.25 | 6.15 | 4.73 | 1.7 | 5.15 | 2.61 | 7.6 / 8.04 |
| | CNNs-v2 | 16.93 | 8.67 | 4.02 | 4.08 | 5.77 | 3.56 | 7.17 / 4.69 |
| | ENet | 22.53 | 5.83 | 18.8 | 2.07 | 8.49 | 3.03 | 10.13 / 7.8 |
| | CNext | 3.97 | 11.62 | 1.1 | 29.97 | 3.14 | 14.75 | 10.76 / 9.86 |
| | DeiT | 23.65 | 12.16 | 7.27 | 5.21 | 32.39 | 9.35 | 15.01 / 9.77 |
| | Swin | 25.2 | 20.21 | 8.93 | 19.54 | 16.16 | 45.31 | 22.56 / 11.3 |
| **Ours ($\mathbf{W}_2^2)^{(1)}$** | CNNs-v1 | 39.65 | 13.01 | 8.27 | 2.44 | 4.89 | 3.16 | 11.9 / 12.91 |
| | CNNs-v2 | 19.0 | 11.35 | 3.82 | 4.51 | 3.74 | 4.19 | 7.77 / 5.69 |
| | ENet | 35.12 | 10.45 | 32.0 | 2.27 | 7.8 | 3.79 | 15.24 / 13.25 |
| | CNext | 3.47 | 12.2 | 0.92 | 25.14 | 2.04 | 15.12 | 9.82 / 8.64 |
| | DeiT | 22.26 | 11.43 | 10.18 | 5.29 | 39.51 | 9.25 | 16.32 / 11.59 |
| | Swin | 20.55 | 17.89 | 8.09 | 17.7 | 13.55 | 49.1 | 21.14 / 13.12 |

reported. Our patch obtains the best transfer results and performs well in a complex setting: scan with defense (see Table 5). This result confirms the potentially harmful behavior of our patch in the real-world.

## 4.3 QUALITATIVE PHYSICAL EXPERIMENTS

In this section, we give some qualitative results concerning the physicality of our attack. We select three objects present in ImageNet-1K (banana, cup, keyboard) and record videos of them when

Table 4: Transfer results on robustified models by LGS defense (Naseer et al., 2019b) (tSuc (%)). Patches are designed on Swin models.

|  |  | CNNs-v1 | CNNs-v2 | Target ENet | CNext | DeiT | Swin | mean / std |
|---|---|---|---|---|---|---|---|---|
|  | Clean | 0.1 | 0.1 | 0.1 | 0.1 | 0.1 | 0.1 | 0.1 / 0 |
|  | GAP (Brown et al., 2017) | 0.72 | 0.87 | 0.35 | 0.78 | 1.13 | 2.34 | 1.03 / 0.63 |
|  | LaVAN (Karmon et al., 2018) | 0.56 | 0.69 | 0.3 | 0.69 | 0.82 | 2.65 | 0.95 / 0.78 |
|  | L2 (Inkawhich et al., 2019) | 4.79 | 6.44 | 1.72 | 7.79 | 4.79 | 13.85 | 6.56 / 3.75 |
| $\lambda = 1.5$ | TnT (Doan et al., 2022) | 0.84 | 0.59 | 0.52 | 0.53 | 0.7 | 0.85 | 0.67 / 0.13 |
|  | Casper et al. (2022) | 0.37 | 0.4 | 0.2 | 0.32 | 0.25 | 0.59 | 0.36 / 0.13 |
|  | TTP (Naseer et al., 2021) | 0.68 | 0.77 | 0.28 | 0.68 | 0.76 | 1.98 | 0.86 / 0.53 |
|  | M3D (Zhao et al., 2023) | 0.83 | 0.81 | 0.36 | 0.77 | 1.17 | 1.17 | 0.85 / 0.27 |
|  | **Ours $(\mathbf{SW}_2^2)_{500}^{(1)}$** | **10.56** | **11.86** | **3.81** | **18.9** | **11.67** | **31.68** | **14.75 / 8.75** |
|  | **Ours $(\mathbf{W}_2^2)^{(1)}$** | **13.23** | **13.4** | **4.37** | **21.42** | **13.84** | **32.08** | **16.39 / 8.58** |

one patch is placed or not next to the object. During the video, patches are moved around the object. Figure 2 shows examples of our patch near objects. In conducted experimentations, all the patches were not able to transfer (tSuc lower than 2%), except for L2 and our patches. The transfer results for the L2 method, our first $((\mathbf{SW}_2^2)_{500}^{(1)})$ and second $((\mathbf{W}_2^2)_{500}^{(1)})$ methods are 9.3%, 23.4% and 29.3% respectively. These results confirm that real-world classifiers can be swayed without explicit knowledge of their architecture or their weights.

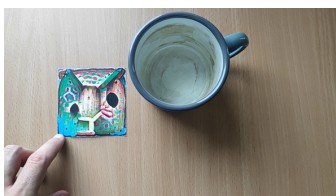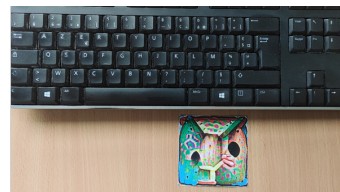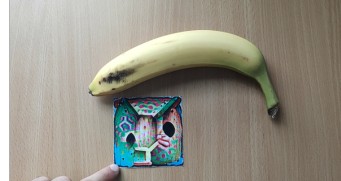

Figure 2: Examples of frame of our APA close to different objects. Our patch is designed to sway networks to output the class bird house.

### 4.4 ABLATION STUDIES

We study the impact of the choice of the targeted layers on the patch transferability. We apply our attack $((\mathbf{W}_2^2)$ version) on different layers for Swin models and report results in Table 6. The last layer of the encoder ($l = l_J$) seems essential to model and close the gap between the corrupted image distribution and the target distribution. It is coherent since this layer is expected to model and separate classes before linear classification. We observe that the multi-layer objective leads to better results as it helps the optimization to converge to a better local minimum, leading to a stronger patch. However, most layers fail to model the targeted distribution correctly. In Appendix I, we propose to study the effect of the power $p$ and the number of slices $K$.

Table 6: Transfer results of digital patches when varying the choice of the targeted layers (tSuc (%)). Patches are designed on Swin models. See Appendix I for details concerning layers.

| $\mathcal{L}$ | $\{l_{J-8}, l_{J-2}, l_J\}$ | $\{l_{J-2}, l_J\}$ | $\{l_{J-2}\}$ | $\{l_J\}$ |
|---|---|---|---|---|
| mean | 14.66 | **23.47** | 17.31 | 21.14 |

## 5 DISCUSSION AND CONCLUSION

This paper presents a distribution-oriented method based on optimal transport for designing APAs. This new method reduces patch overfitting to the source architecture and strengthens its transferability to Convolutional Neural Networks and Transformer architectures. When designed on Swin models, our patch is the only one capable of strongly fooling multiple architectures from different model families, even when the model robustness has been enhanced by a defense mechanism. Hybrid and physical experiments illustrate that our attack can disturb real-world classifiers without any knowledge of the system.

## 6 ETHICS STATEMENT

This paper presents how to make attacks potentially harmful to real-life deep networks. Ignoring the existence of attacks like the one presented in this work leaves systems with a false sense of security and may be disastrous to numerous applications like autonomous vehicles. Research promotes transparency and fosters a proactive approach to addressing potential vulnerabilities. We hope that our work empowers individuals and organizations to take necessary precautions, ultimately leading to a safer and more secure AI landscape.

Although this paper is just one snapshot of the attack/defense race, we want to summarize some recommendations based on our experiments.

- Despite Swin being a state-of-the-art model, we advise against relying on it for critical computer vision functions. Indeed, an attacker using our method would design its patch on Swin models to maximize its transferability across the different network families. However, as we show, patches designed on Swin models are the most critical for Swin models.

- Conversely, using either of ConvNext or AT model seems a good shot: AT models combined with defense are quite resilient to patch not designed on them with the drawback of a moderate initial performance, and, ConvNext are the best trade-off today (good initial performance and moderate loss of performance against an attack even when designed on ConvNext).

## ACKNOWLEDGMENTS

This work has been supported by the French government under the "France 2030" program, as part of the SystemX Technological Research Institute within the Confiance.ai project.
This work was granted access to the HPC resources of IDRIS under the allocation 20AD-011014245 made by GENCI.

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

# Appendices

## A  IMPLEMENTATION DETAILS

Our method training routine uses the PyTorch library (Paszke et al., 2019). For the training of each patch on medium and large models we consider a single NVIDIA V100-32G or a single NVIDIA A100 respectively. To train patches on smaller NVIDIA cards we should reduce the batch size.

When not specified, patches are designed to target one the following classes: salamander, starfish, bird house, bullfrog, pinwheel, mongoose, brown bear, accordion and common iguana.

We use Expectation over Transformations (EoT (Eykholt et al., 2018)) to obtain a more physically realizable patch, similarly to prior work on APAs (Brown et al., 2017; Lee & Kolter, 2019; Casper et al., 2022). For all the methods (GAP (Brown et al., 2017), LaVAN (Karmon et al., 2018), L2 (Inkawhich et al., 2019) and ours), during training, we randomly rotate the patch up to five degrees for the x and y-axis and up to 10 degrees for the z-axis. We also randomly scale the patch between $70 \times 70$ to $110 \times 110$ pixels, adjust patch brightness between $[-0.1, 0.1]$ and patch blur between $[0.8, 1.2]$, and apply normal noise of magnitude 0.1 on the patch. Patches are randomly translated in the image but not in the center.

To control the balance between the adversarial loss and the total variation loss, the gradient of each loss is computed individually, normalized, and combined using a weighted sum. Following Nesti et al. (2022) we choose $w_{adv} = 1$ and $w_{TV} = 0.1$ where $w_{adv}$ is the weight for the adversarial loss and $w_{TV}$ is the weight for the TV loss.

**Computation time.**  We measure and report the computation time of each method in Table 7. This Table reports the averaged computational time for the different methods. Our method has a similar computational time as other methods. This result may be counterintuitive as OT losses are known to be slow, but in our setting, the number of samples is low. The M3D method (Zhao et al., 2023) is

much slower than other methods. It is coherent, this method trains alternatively the patch and two models in a min-max game.

Table 7: Computational time of the different methods to obtain a fully optimized patch (minutes). Times are averaged over ten optimization runs. Each run is launched on the same setup composed by a single NVIDIA A100.

| Method | Time |
|---|---|
| GAP (Brown et al., 2017) | 20 |
| LaVAN (Karmon et al., 2018) | 30 |
| L2 (Inkawhich et al., 2019) | 20 |
| TnT (Doan et al., 2022) | 30 |
| Casper et al. (2022) | 35 |
| TTP (Naseer et al., 2021) | 30 |
| M3D (Zhao et al., 2023) | 66 |
| **Ours** $(\mathbf{SW}_2^2)_{500}^{(1)}$ | 19 |
| **Ours** $(\mathbf{W}_2^2)^{(1)}$ | 20 |

## B  FEATURE POINT METHOD INSTABILITY

To measure the stability of the L2 method (Inkawhich et al., 2019), we launch the optimization for three randomly selected target points. Patches are designed to sway ResNet50-v1 or Swin-T to output the class Australian terrier. Figure 3 plots the learning curves and the resulting patches for our distribution-based approach for Resnet50-v1 and Swin-T, respectively. Figure 4 and 5 plot the learning curves and the resulted patches of the L2 method for Resnet50-v1 and Swin-T, respectively. These four graphs show that our method is the easiest to optimize and is more robust to optimization artifacts. For the Swin-T model, the optimization for the L2 method becomes noisy. Table 8 reports the transfer results of the obtained patches from previous figures. Although the optimization has converged for the first target of the L2 method for ResNet50-v1, the obtained patch is harmless. Even if the APA works, its attacking capacity depends on the considered target point. For example, the mean transferability on Swin-T can decreased by a factor four. In general, our distribution-oriented approach outperforms the L2 method.

Table 8: Transfer results between categories of models (tSuc (%)) for the L2 method and for our distribution-oriented method. Three different target points are evaluated for the L2 method. Results are for the source model ResNet50-V1 and Swin-T, for the class Australian terrier and for patches of size $60 \times 60$. Patches are placed randomly in the image but not at the center of images.

| Source | Method | | CNNs-v1 | CNNs-v2 | ENet | CNext | DeiT | Swin | AT | mean / std |
|---|---|---|---|---|---|---|---|---|---|---|
| ResNet50-v1 | L2 (Inkawhich et al., 2019) | Target 1 | 0.1 | 0.1 | 0.1 | 0.1 | 0.1 | 0.1 | 0.1 | 0.1 / 0 |
| | | Target 2 | 36.64 | 2.52 | 9.35 | 0.52 | 3.59 | 0.5 | 3.71 | 8.12 / 12 |
| | | Target 3 | 43.83 | 4.18 | 8.82 | 0.75 | 4.98 | 0.58 | 6.09 | 9.89 / 14.1 |
| | **Ours** | | 43.34 | 4.76 | 8.75 | 0.92 | 6.46 | 0.63 | 4.68 | **9.94 / 13.9** |
| Swin-T | L2 (Inkawhich et al., 2019) | Target 1 | 4.12 | 1.18 | 2.41 | 0.23 | 1.83 | 1.9 | 0.39 | 1.72 / 7.8 |
| | | Target 2 | 26.97 | 7.36 | 4.65 | 3.9 | 7.2 | 6.13 | 1.92 | 8.3 / 7.8 |
| | | Target 3 | 0.17 | 0.11 | 0.12 | 0.1 | 0.1 | 0.07 | 0.1 | 0.11 / 0.02 |
| | **Ours** | | 50.77 | 12.54 | 14.2 | 7.08 | 13.64 | 8.19 | 5.94 | **16.05 / 14.5** |

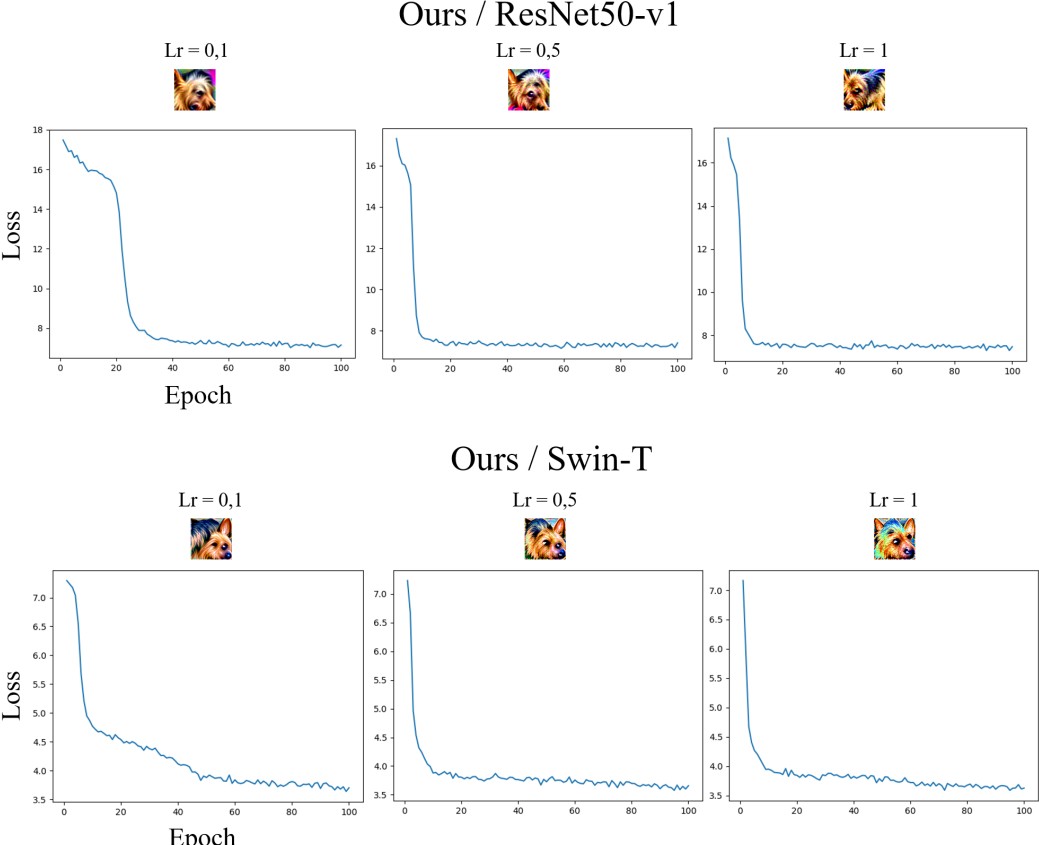

Figure 3: Learning curves and resulted patches of our distribution-oriented method. The optimization is run for three different learning rate. The source model is ResNet50-v1 or Swin-T and the targeted class is Australian terrier.

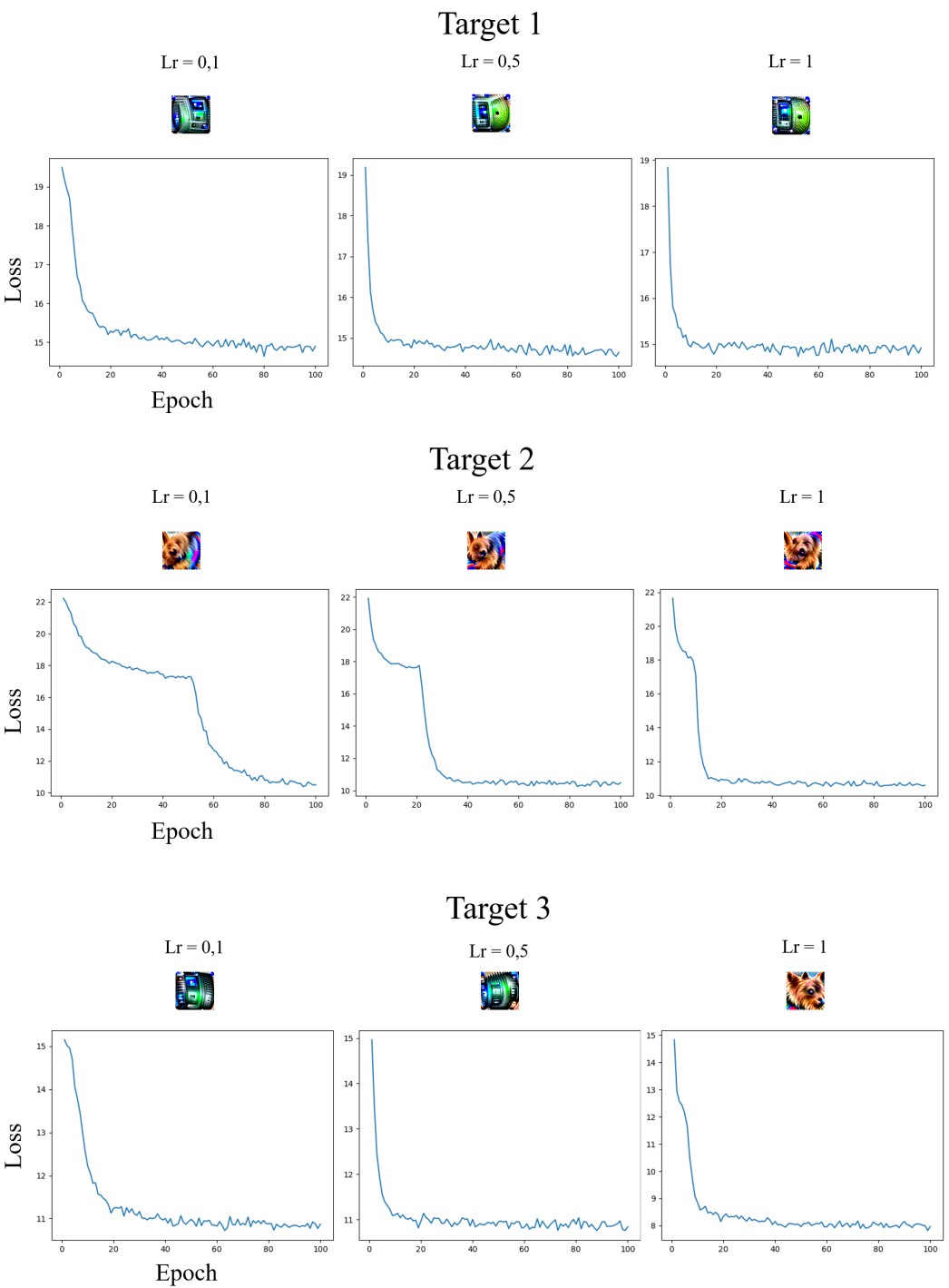

Figure 4: Learning curves and resulted patches of the L2 method for different targeted points. For each targeted point the optimization is run for three different learning rate. The source model is ResNet50-v1 and the targeted class is Australian terrier.

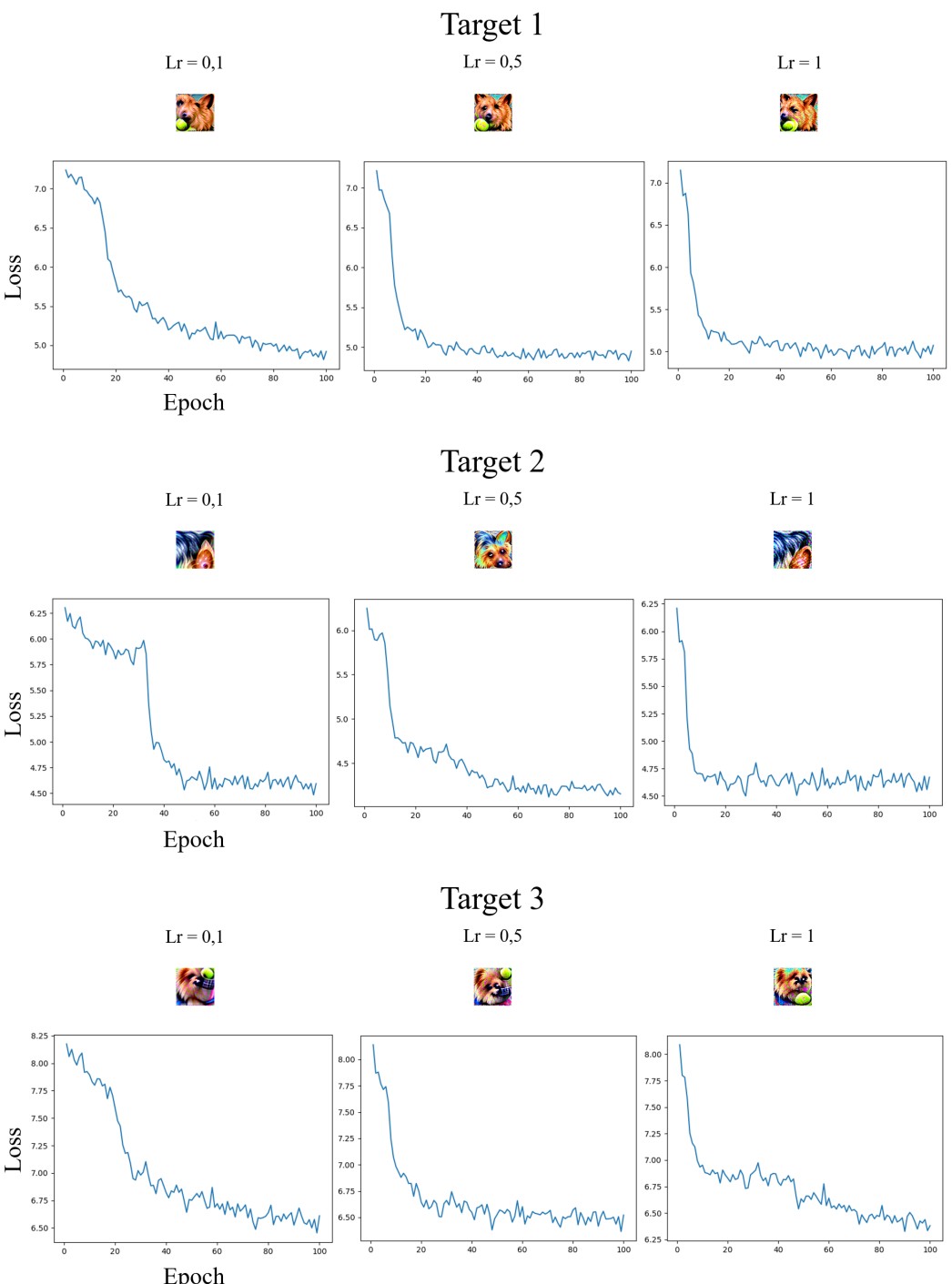

Figure 5: Learning curves and resulted patches of the L2 method for different targeted points. For each targeted point the optimization is run for three different learning rate. The source model is Swin-T and the targeted class is Australian terrier.

## C    FEATURE POINT METHOD GENERALIZATION

We provide in this section a proof that the exact 2-Wasserstein distance coincide with the L2-based method Inkawhich et al. (2019) when the source distribution is uniformly distributed and the targeted distribution is supported by a unique point. We recall that

$$\mathbf{W}_2^2(\mu, \nu) = \inf_{\pi \in \Pi(\mu,\nu)} \int_{\mathbb{R}^d \times \mathbb{R}^d} ||x - y||^2 \mathrm{d}\pi(x, y), \tag{6}$$

defines the 2-Wasserstein distance. This distance can be interpreted through a probabilistic point of view. If we name $(X, Y)$ a couple of random variables over $\mathbb{R}^d \times \mathbb{R}^d$ with $X \sim \mu$, $Y \sim \nu$ and $(X, Y) \sim \pi \in \Pi(\mu, \nu)$, we can write

$$\mathbf{W}_2^2(\mu, \nu) = \min_{(X,Y)} \mathbb{E}_{(X,Y)} \left[ ||X - Y||^2 \right]. \tag{7}$$

If we suppose that the target distribution is composed by a unique point, *i.e.*, $\nu = \delta_y$, then we have

$$\mathbf{W}_2^2(\mu, \nu) = \min_X \mathbb{E}_X \left[ ||X - y||^2 \right]. \tag{8}$$

In our problem we have empirical distribution based on samples, we name $\hat{\mu}_n$ and $\hat{\nu}_m$ the empirical distributions of $\mu$ based on $n$ samples and $\nu$ based on $m$ samples respectively. We suppose that each sample from each distribution is uniformly distributed, *i.e.*, $\hat{\mu}_n = \frac{1}{n} \sum_{i=1}^n \delta_{x_i}$ and $\hat{\nu}_m = \frac{1}{m} \sum_{j=1}^m \delta_{y_j}$, where $\delta$ is the Kronecker symbol. The estimated 2-Wasserstein distance is

$$\mathbf{W}_2^2(\hat{\mu}_n, \hat{\nu}_m) = \min_{\pi \in \Pi(\mu,\nu)} \sum_{i=1}^n \sum_{j=1}^m \pi_{ij} ||x_i - y_j||^2. \tag{9}$$

If we suppose that the target distribution is composed by a unique point, *i.e.*, $\hat{\nu} = \delta_y$, then we have

$$\mathbf{W}_2^2(\hat{\mu}_n, \hat{\nu}) = \frac{1}{n} \sum_{i=1}^n ||x_i - y||^2. \tag{10}$$

which is equal to the L2-based criterion. Minimizing with respect to the 2-Wasserstein is equivalent to consider the L2-based criterion (Inkawhich et al., 2019). As a result, our method includes and generalizes the L2-based method.

## D    BENEFITS OF OPTIMAL TRANSPORT

Optimal transport-based losses (both exact and sliced) has the following advantages:

- OT losses take into account the underlying metric space (through the cost matrix) on which the probability distributions are defined,
- for non-overlapping distributions such as ours, the Kullback-Leibler divergence is infinite.

To illustrate the first point, we consider the toy example shown in Figure 6. We define four different one-dimensional distributions supported here by five points. We compute the 1-Wasserstein distance and the KL divergence between the red and the blue distributions for each column (results are shown between graphs). The blue mass has been moved near the first point from right to left. The 1-Wasserstein distance captures this mass shift, while the KL divergence does not and remains constant. This toy example highlights that OT losses capture the underlying geometry on which distributions are defined. More details concerning the advantages of OT over other methods can be found in (Arjovsky et al., 2017) (Part 2: Different Distances).

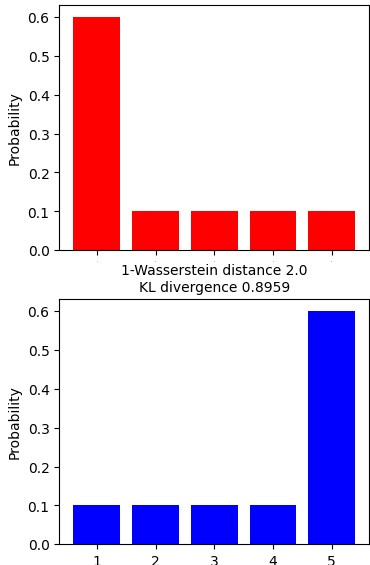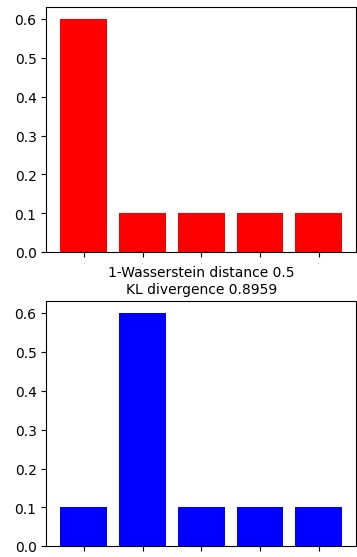

Figure 6: Example of distributions defined on five points with different mass values. The 1-Wasserstein distance and the KL divergence is computed between the red and the blue distribution for each column.

## E  MODEL ROBUSTNESS AND PATCH POSITION

In this section, we evaluate the robustness of models according to the patch position in images. We consider the same families of models as before. We define nine patch positions and measure the patch transferability when the patch is fixed at one of these positions. Figure 7 represents the nine patch positions. We regroup these positions into three categories: Corner, Cross, and Center. We measure the patch transferability for a patch of size $40 \times 40$ ($\approx 3\%$ image size). Results are averaged over methods (GAP (Brown et al., 2017), LaVAN (Karmon et al., 2018), L2 (Inkawhich et al., 2019) and ours), classes, and categories of patch position. Table 9 reports the patch transferability according to its position. CNNs-v1 models are much more biased by the center of images than other network families. The accuracy of CNNs-v1 drops by a factor of 14 % when the patch is moved to corners to the center of images. This effect is not entirely due to the occluding of the object of interest since the patch is very small. Very recent families of networks (CNext and Swin models) are the more balanced networks in using context in images. For these models, the accuracy is nearly the same when the patch is placed in either corners or the center. To measure the actual efficiency of patches and to not occlude the object of interest in the case of large patches, its patches may not be placed in the center of images.

Table 9: Transfer results according to the categories of patch position (Accuracy (%)). Results are averaged over methods, over classes, over patch positions and are for patches of size $40 \times 40$.

|  | | | | Target | | | |
|---|---|---|---|---|---|---|---|
|  | CNNs-v1 | CNNs-v2 | ENet | CNext | DeiT | Swin | AT |
| Clean | 74.90 | 77.63 | 80.15 | 82.42 | 77.81 | 82.43 | 65.44 |
| Position | | | | | | | |
| Corner | 71.07 | 76.57 | 78.85 | 81.97 | 76.33 | 81.43 | 64.24 |
| Cross | 67.65 | 75.36 | 77.71 | 81.66 | 75.66 | 81.44 | 62.44 |
| Center | 61.52 | 72.01 | 74.23 | 80.72 | 73.94 | 80.71 | 57.06 |

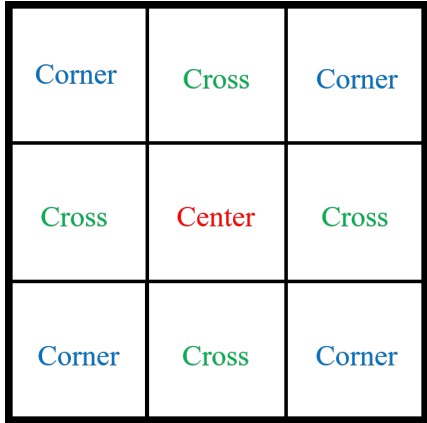

Figure 7: Illustration of the categories of patch positions.

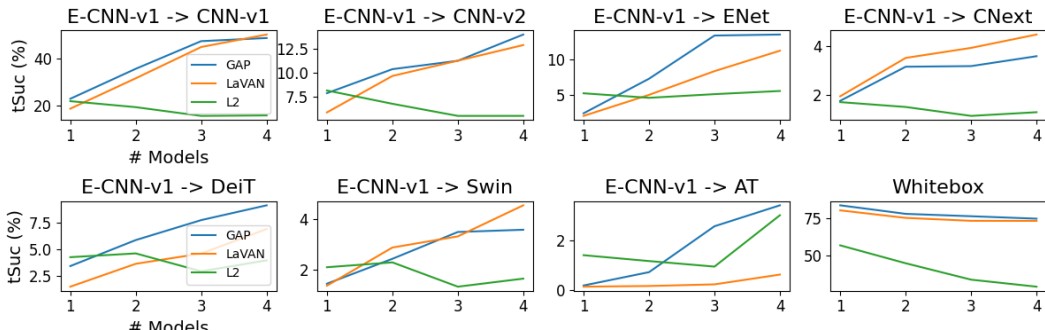

Figure 8: Transfer and whitebox results for patches built on an ensemble of models (tSuc (%)). Results are averaged over classes and over patch sizes. Patches are placed randomly in the image but not at the center of images.

## F  ENSEMBLE METHODS

Ensemble methods train a single patch across an ensemble of models simultaneously. We determine if an attacker building his attack on an ensemble of CNN-v1 models can significantly increase its attacking performance on CNext or Swin models. We consider the following ordered list of models E-CNN-v1 = {ResNet50/34/18-v1, DenseNet121} in which networks are added to the ensemble in this order. Figure 8 plots the targeted success rate (tSuc) as a function of the number of models in the ensemble. Even with the largest ensemble of four models, patches failed to significantly increase their transferability performances on CNext and Swin models. This result confirms that an attacker expecting to sway all the models uniformly should design his attack on Swin models using our methodology. Figure 8 also shows that the feature point method becomes unstable with the increased number of models in the ensemble.

## G  TRANSFERABILITY ON ADVERSARIALLY TRAINED MODELS

In this section, we study the robustness of Adversarially Trained (AT) models. We consider two scenarios: when the patch is learned on AT models and when not. To strongly transfer on an AT model, the patch must be designed on an AT model (Table 10). None of the other source models can show good transferability results when applied to AT models. These results suggest that AT models learn different representations than other networks. From Table 11, we see that the GAP method (Brown et al., 2017) and our method are the best procedures to design a patch to target an AT model.

Table 10: Transfer results between categories of models (tSuc (%)). Results are averaged over classes and over patch sizes. Patches are designed using our method $(\mathbf{W}_2^2)^{(1)}$.

| | CNNs-v1 | CNNs-v2 | ENet | Target CNext | DeiT | Swin | AT | mean / std |
|---|---|---|---|---|---|---|---|---|
| Clean Source | 0.1 | 0.1 | 0.1 | 0.1 | 0.1 | 0.1 | 0.1 | 0.1 / 0 |
| CNNs-v1 | 39.65 | 13.01 | 8.27 | 2.44 | 4.89 | 3.16 | 0.82 | 10.32 / 12.56 |
| CNNs-v2 | 19.0 | 11.35 | 3.82 | 4.51 | 3.74 | 4.19 | 0.45 | 6.72 / 5.86 |
| ENet | 35.12 | 10.45 | 32.0 | 2.27 | 7.8 | 3.79 | 3.49 | 13.56 / 12.94 |
| CNext | 3.47 | 12.2 | 0.92 | 25.14 | 2.04 | 15.12 | 0.16 | 8.44 / 8.69 |
| DeiT | 22.26 | 11.43 | 10.18 | 5.29 | 39.51 | 9.25 | 5.08 | 14.72 / 11.43 |
| Swin | 20.55 | 17.89 | 8.09 | 17.7 | 13.55 | 49.1 | 0.72 | 18.23 / 14.09 |
| AT | 39.75 | 10.69 | 17.35 | 3.51 | 19.87 | 5.31 | 38.95 | 19.35 / 13.77 |

Table 11: Transfer results between categories of models (tSuc (%)). Results are averaged over classes and over patch sizes. Patches are placed randomly in the image but not at the center of images.

| | CNNs-v1 | CNNs-v2 | ENet | Target CNext | DeiT | Swin | AT | mean / std |
|---|---|---|---|---|---|---|---|---|
| Clean Method | 0.1 | 0.1 | 0.1 | 0.1 | 0.1 | 0.1 | 0.1 | 0.1 / 0 |
| GAP (Brown et al., 2017) | 43.05 | 11.67 | 16.7 | 3.35 | 20.09 | 5.23 | 39.17 | 19.98 / 14.51 |
| LaVAN (Karmon et al., 2018) | 37.27 | 10.94 | 14.08 | 3.43 | 18.18 | 5.21 | 29.96 | 17.018 / 11.64 |
| L2 (Inkawhich et al., 2019) | 6.78 | 1.86 | 2.23 | 0.59 | 4.39 | 1.1 | 8.35 | 3.618 / 2.77 |
| TnT (Doan et al., 2022) | 3.71 | 1.33 | 1.41 | 0.8 | 2.61 | 0.85 | 8.03 | 2.688 / 2.39 |
| Casper et al. (2022) | 5.83 | 1.38 | 2.74 | 0.54 | 7.55 | 0.97 | 13.91 | 4.78 / 4.48 |
| TTP (Naseer et al., 2021) | 35.25 | 9.45 | 13.75 | 2.91 | 17.92 | 4.69 | 35.17 | 17.028 / 12.43 |
| M3D (Zhao et al., 2023) | 6.24 | 5.61 | 3.45 | 0.82 | 1.82 | 1.12 | 2.53 | 3.088 / 1.98 |
| **Ours** $(\mathbf{SW}_2^2)^{(1)}$ | 22.52 | 5.22 | 8.05 | 2.17 | 11.51 | 3.22 | 21.57 | 10.618 / 7.79 |
| **Ours** $(\mathbf{W}_2^2)^{(1)}$ | 39.75 | 10.69 | 17.35 | 3.51 | 19.87 | 5.31 | 38.95 | 19.35 / 13.77 |

# H ROBUSTNESS ACCORDING TO PHYSICAL TRANSFORMATIONS

In this section, we measure the robustness of patches according to physical transformations. We evaluate the L2 (Inkawhich et al., 2019), our exact Wasserstein $(\mathbf{W}_2^2)^{(1)}$ and Sliced-Wasserstein $(\mathbf{SW}_2^2)_{500}^{(1)}$ patches as they are the only to transfer in the easiest scenario, *i.e.*, without patch rotation, medium brightness and small distance patch-camera (Section 4.3 of the main article). Patch transferability is measured according to z-axis rotations (rotations in the image plane), variation of light (low and high) and distance between camera and the object (the patch is placed near the object). Results are reported in Table 12 and Figure 9. Our patches transfer even in the worst-case scenario (far from the camera or when rotated), while other patches do not. This indicates that our patches may be critical in real-world scenarios. Globally, our method produces patches with better transferability than other methods.

Table 12: Transfer results according to rotations and variation of light (tSuc %). Patches are designed to sway networks to output the class bird house. Patches are printed and placed in the real-world near a cup. Results are averaged over video frames and over all the networks.

| Method | z-axis rotations -45° | 0° | 45° | Variation of light Low | High |
|---|---|---|---|---|---|
| L2 (Inkawhich et al., 2019) | 0.8 | 5.7 | 0.23 | 4.4 | 5.7 |
| **Ours** $(\mathbf{SW}_2^2)_{500}^{(1)}$ | 6.1 | 11.5 | 6.53 | 12 | 11.5 |
| **Ours** $(\mathbf{W}_2^2)^{(1)}$ | **7.1** | **14.8** | **7.05** | **12.6** | **14.8** |

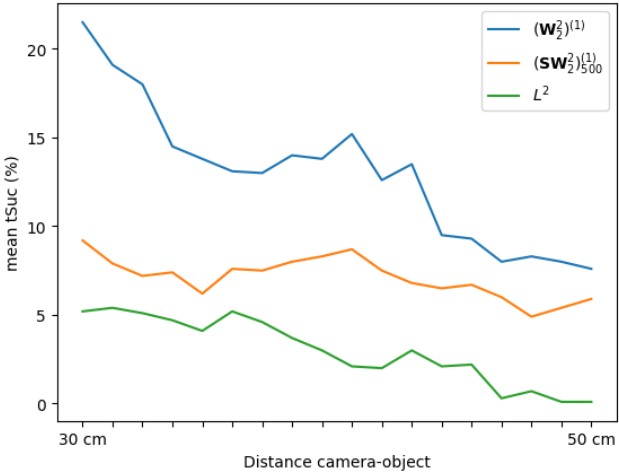

Figure 9: Transfer results as a function of the distance camera-object. Patches are designed to sway networks to output the class bird house. Patches are printed and placed in the real-world near a cup. Results are averaged over video frames and over all the networks.

## I    ABLATION STUDIES

In this section, we study the effect of our method hyper-parameters. We solve the exact and the sliced Wasserstein distance for $p \in \{1, 2\}$ and report the results in Table 13. This Table shows that both values of $p$ lead to the same transferability. To penalize higher feature values, we set the value of $p = 2$.

We launch the Sliced-Wasserstein distance (**SW**) for the following number of projections: $K \in \{500, 1000, 5000, 10000, 50000\}$. There is no clear advantage to considering many projections (Table 16). The best transferability results are obtained with $K = 500$.

We now study the effect of the number of attacked layers ($N$). In Table 14, we report the transferability results according to different numbers of targeted layers. We obtain better results for the exact Wasserstein distance when considering multiple layers. We observe that it helps the optimization to converge to a better local minimum, leading to stronger patches. For the Sliced-Wasserstein distance, targeting multiple layers seems counterproductive. Table 15 details the result presented in the article on the choice of the essential layer to target. The last layer of the encoder ($l = l_J$) seems essential to model and close the gap between the two distributions and, particularly, for the Sliced-Wasserstein distance.

To evaluate the data dependency of our method, we create different targeted distributions by changing the number of points which compose it ($m = 1, 2, 10, 100, 300, 600, 900$). We launch the optimization of patches for five different sampling seeds and three different classes. We consider the Swin-T model as the source model. We evaluate patches using the same procedure explained in the main article (Section 4). We report the results of the three runners-up baselines (GAP, LaVAN and TTP). As these methods do not consider distributions, they correspond to straight lines in the figure. From Figure 11 we see that the average targeted success rate (tSuc) increases with respect to the number of target samples. When considering multiple points, our method leads to better transfer results and is more stable than the L2-based method (see B). Our method performs better than decision-boundary-based methods (GAP, LaVAN and TTP). However, we would like to emphasize that our method requires multiple images of the target class to overcome the limitations of the L2-based approach (see Appendix B). This data dependency is a practical limitation of our method. This practical limitation may be simply leveraged by considering the training data of the source model when available.

Table 13: Transfer results according to the power $p$ (tSuc (%)). Results are averaged over classes and over patch sizes. Patches are designed on Swin-T.

| | CNNs-v1 | CNNs-v2 | ENet | Target CNext | DeiT | Swin | mean / std |
|---|---|---|---|---|---|---|---|
| Clean $p$ | 0.1 | 0.1 | 0.1 | 0.1 | 0.1 | 0.1 | 0.1 / 0 |
| $(\mathbf{W}_2^2)^{(1)}$ | 25.62 | 19.88 | 10.96 | 18.84 | 13.28 | 55.67 | 24.04 / 14.91 |
| $(\mathbf{W}_1^1)^{(1)}$ | 26.37 | 20.99 | 10.86 | 19.56 | 13.3 | 56.98 | 24.68 / 15.31 |
| $(\mathbf{SW}_2^2)_{10000}^{(1)}$ | 27.82 | 20.22 | 11.29 | 18.6 | 16.66 | 41.43 | 22.67 / 9.72 |
| $(\mathbf{SW}_1^1)_{10000}^{(1)}$ | 28.74 | 22.72 | 11.24 | 19.89 | 16.07 | 43.13 | 23.63 / 10.26 |

Table 14: Transfer results according to the number of targeted layers ($N$) (tSuc (%)). Results are averaged over classes and over patch sizes. Patches are designed on the Swin family. Layers $l_{J-8}$ and $l_{J-2}$ correspond to the second and third block of Swin models (which are composed by four blocks in total).

| | | CNNs-v1 | CNNs-v2 | ENet | Target CNext | DeiT | Swin | mean / std |
|---|---|---|---|---|---|---|---|---|
| | Clean ($N$) | 0.1 | 0.1 | 0.1 | 0.1 | 0.1 | 0.1 | 0.1 / 0 |
| $(\mathbf{W}_2^2)^{(N)}$ | $\{l_J\}$ | 20.55 | 17.89 | 8.09 | 17.7 | 13.55 | 49.1 | 21.14 / 13.12 |
| | $\{l_{J-2}, l_J\}$ | 24.87 | 20.38 | 9.59 | 19.77 | 17.77 | 48.42 | 23.47 / 12.06 |
| | $\{l_{J-8}, l_{J-2}, l_J\}$ | 19.14 | 12.45 | 7.52 | 10.56 | 13.55 | 24.75 | 14.66 / 14.66 |
| $(\mathbf{SW}_2^2)^{(N)}$ | $\{l_J\}$ | 25.26 | 18.7 | 9.19 | 17.27 | 15.32 | 44.11 | 21.64 / 11.11 |
| | $\{l_{J-2}, l_J\}$ | 24.22 | 18.26 | 8.25 | 15.27 | 17.47 | 34.5 | 19.66 / 8.14 |
| | $\{l_{J-8}, l_{J-2}, l_J\}$ | 15.94 | 10.23 | 6.35 | 8.6 | 12.43 | 17.35 | 11.82 / 3.89 |

Table 15: Transfer results according to targeted layer in the single targeted layer setting (tSuc (%)). Results are averaged over classes and over patch sizes. Patches are designed on the Swin family. Layers $l_{J-8}$ and $l_{J-2}$ correspond to the second and third block of Swin models (which are composed by four blocks in total).

| | | CNNs-v1 | CNNs-v2 | ENet | Target CNext | DeiT | Swin | mean / std |
|---|---|---|---|---|---|---|---|---|
| | Clean $\mathcal{L}$ | 0.1 | 0.1 | 0.1 | 0.1 | 0.1 | 0.1 | 0.1 / 0 |
| $(\mathbf{W}_2^2)^{(1)}$ | $l_{J-2}$ | 17.02 | 15.03 | 6.59 | 14.32 | 12.55 | 38.35 | 17.31 / 9.95 |
| | $l_J$ | 20.55 | 17.89 | 8.09 | 17.7 | 13.55 | 49.1 | 21.14 / 13.12 |
| $(\mathbf{SW}_2^2)^{(1)}$ | $l_{J-8}$ | 0.3 | 0.19 | 0.19 | 0.14 | 0.17 | 0.2 | 0.2 / 0.05 |
| | $l_{J-2}$ | 15.39 | 11.2 | 5.2 | 9.08 | 13.37 | 20.44 | 12.45 / 4.81 |
| | $l_J$ | 25.26 | 18.7 | 9.19 | 17.27 | 15.32 | 44.11 | 21.64 / 11.11 |

# J  DECISION BOUNDARY-BASED METHODS OVERFITTING

In this section, we conduct an additional experiment to support that decision boundary-based methods learn a patch that tends to overfit on the source model classifier. For this purpose, we consider the transfer not between 2 different models but between 2 models sharing the same encoder but different classifiers. We select from the different methods patches trained to attack the source model Swin-T (Liu et al., 2021). On top of this Swin-T encoder, we train a new linear classifier from scratch on the ImageNet train set (Deng et al., 2009). This new linear classifier reaches the same level of clean accuracy as the previous classifier (from Pytorch (Paszke et al., 2019)) while being different. We measured the patch performance when targeting this new network (same encoder, different linear classifier). As expected, the transferability of decision boundary-based patches drops drastically (nearly by half) while our patches transferability remains almost the same.

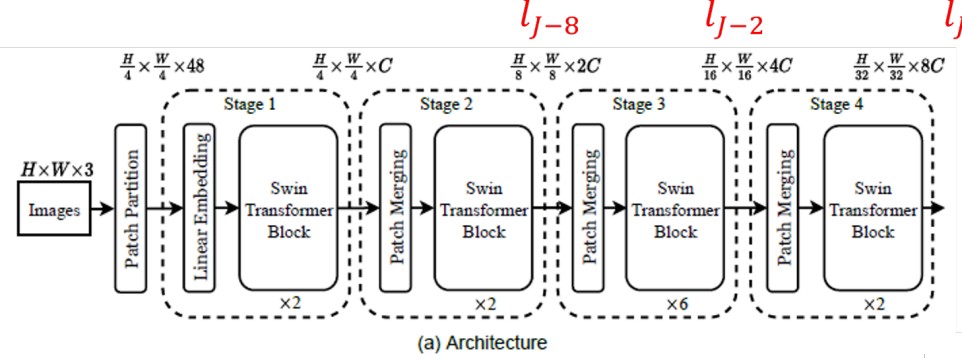

Figure 10: Figure from (Liu et al., 2021). In red are displayed the targeted layers consider in the article.

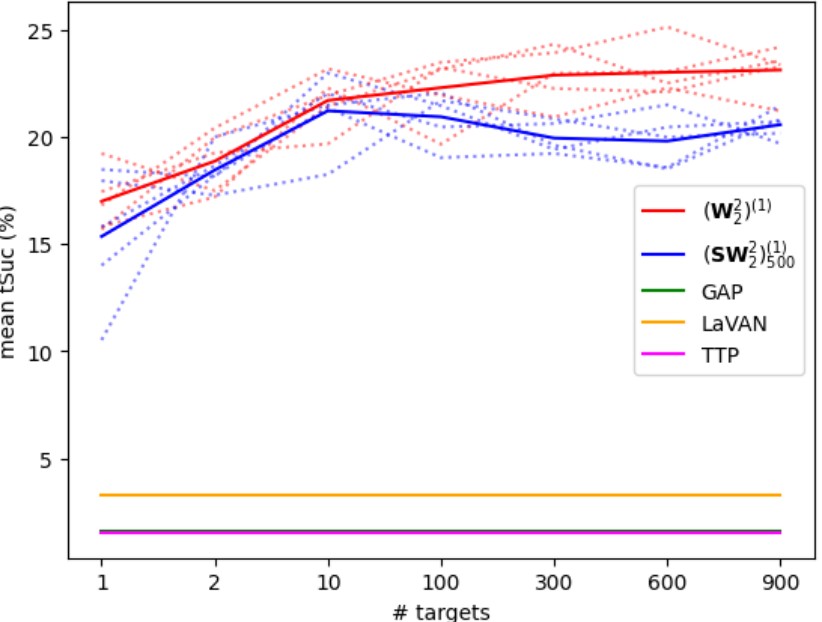

Figure 11: Transfer results as a function of the number of targets points supported in the target distribution (mean tSuc (%)). Each dotted line correspond to a different sampling of points to create the target distribution. The solid line is the average of the five dotted lines. Patches are designed on the Swin-T source model. Results are averaged over three classes, over patch sizes and over all the targeted networks.

Table 16: Transfer results (tSuc (%), higher is better attack) between categories of models. Results are averaged over classes and over patch sizes. Patches are placed randomly in the image without object overlapping. Physical transformations (*e.g.*, noise, rotations) are applied to patches. Control stands for inserting a real object of the corresponding class as a patch.

| Method | Source | CNNs-v1 | CNNs-v2 | ENet | CNext | DeiT | Swin | mean / std |
|---|---|---|---|---|---|---|---|---|
| | Clean | 0.1 | 0.1 | 0.1 | 0.1 | 0.1 | 0.1 | 0.1 / 0 |
| Control | | 2.85 | 1.59 | 0.86 | 0.54 | 1.57 | 0.93 | 1.39 / 0.75 |
| $(\mathbf{SW}_2^2)_{500}^{(1)}$ | CNNs-v1 | 25.25 | 6.15 | 4.73 | 1.7 | 5.15 | 2.61 | 7.6 / 8.04 |
| | CNNs-v2 | 16.93 | 8.67 | 4.02 | 4.08 | 5.77 | 3.56 | 7.17 / 4.69 |
| | ENet | 22.53 | 5.83 | 18.8 | 2.07 | 8.49 | 3.03 | 10.13 / 7.8 |
| | CNext | 3.97 | 11.62 | 1.1 | 29.97 | 3.14 | 14.75 | 10.76 / 9.86 |
| | DeiT | 23.65 | 12.16 | 7.27 | 5.21 | 32.39 | 9.35 | 15.01 / 9.77 |
| | Swin | 25.2 | 20.21 | 8.93 | 19.54 | 16.16 | 45.31 | 22.56 / 11.3 |
| $(\mathbf{SW}_2^2)_{1000}^{(1)}$ | CNNs-v1 | 26.38 | 6.13 | 5.59 | 1.96 | 5.85 | 2.8 | 8.12 / 8.32 |
| | CNNs-v2 | 16.88 | 8.82 | 3.97 | 3.89 | 5.8 | 3.53 | 7.15 / 4.71 |
| | ENet | 23.56 | 6.45 | 19.18 | 2.25 | 8.55 | 3.01 | 10.5 / 8.07 |
| | CNext | 4.44 | 12.07 | 1.14 | 33.22 | 3.24 | 15.3 | 11.57 / 10.89 |
| | DeiT | 22.77 | 11.97 | 7.68 | 5.36 | 35.25 | 9.2 | 15.37 / 10.48 |
| | Swin | 24.2 | 19.01 | 8.94 | 17.73 | 15.89 | 44.53 | 21.72 / 11.16 |
| $(\mathbf{SW}_2^2)_{5000}^{(1)}$ | CNNs-v1 | 26.4 | 6.11 | 5.37 | 1.83 | 5.2 | 2.65 | 7.93 / 8.4 |
| | CNNs-v2 | 14.49 | 8.35 | 3.73 | 3.64 | 5.89 | 3.24 | 6.55 / 3.96 |
| | ENet | 27.44 | 7.04 | 19.85 | 2.16 | 8.88 | 3.12 | 11.42 / 9.2 |
| | CNext | 4.52 | 13.79 | 1.18 | 31.54 | 3.18 | 16.4 | 11.77 / 10.45 |
| | DeiT | 24.14 | 12.89 | 8.37 | 5.02 | 36.29 | 9.17 | 15.98 / 10.9 |
| | Swin | 24.02 | 19.69 | 9.53 | 17.97 | 15.06 | 44.74 | 21.83 / 11.15 |
| $(\mathbf{SW}_2^2)_{10000}^{(1)}$ | CNNs-v1 | 25.73 | 6.25 | 5.51 | 1.86 | 5.75 | 2.67 | 7.96 / 8.11 |
| | CNNs-v2 | 18.38 | 10.46 | 4.19 | 4.73 | 6.15 | 4.01 | 7.99 / 5.14 |
| | ENet | 24.49 | 6.6 | 20.26 | 2.14 | 8.64 | 2.98 | 10.85 / 8.52 |
| | CNext | 2.92 | 9.34 | 0.92 | 23.33 | 2.9 | 12.18 | 8.6 / 7.68 |
| | DeiT | 23.87 | 12.22 | 7.57 | 4.89 | 36.3 | 9.34 | 15.7 / 11.01 |
| | Swin | 23.68 | 18.08 | 8.92 | 17.95 | 15.42 | 44.61 | 21.44 / 11.24 |
| $(\mathbf{SW}_2^2)_{50000}^{(1)}$ | CNNs-v1 | 26.16 | 6.16 | 5.4 | 1.89 | 5.32 | 2.7 | 7.94 / 8.29 |
| | CNNs-v2 | 13.67 | 8.71 | 3.09 | 4.0 | 4.67 | 3.4 | 6.26 / 3.8 |
| | ENet | 25.66 | 6.06 | 20.4 | 2.11 | 8.73 | 2.99 | 10.99 / 8.91 |
| | CNext | 3.06 | 10.97 | 0.95 | 27.34 | 3.34 | 16.73 | 10.4 / 9.31 |
| | DeiT | 23.95 | 11.84 | 8.65 | 4.6 | 35.72 | 8.58 | 15.56 / 10.86 |
| | Swin | 25.26 | 18.7 | 9.19 | 17.27 | 15.32 | 44.11 | 21.64 / 11.11 |
| $(\mathbf{W}_2^2)^{(1)}$ | CNNs-v1 | 39.65 | 13.01 | 8.27 | 2.44 | 4.89 | 3.16 | 11.9 / 12.91 |
| | CNNs-v2 | 19.0 | 11.35 | 3.82 | 4.51 | 3.74 | 4.19 | 7.77 / 5.69 |
| | ENet | 35.12 | 10.45 | 32.0 | 2.27 | 7.8 | 3.79 | 15.24 / 13.25 |
| | CNext | 3.47 | 12.2 | 0.92 | 25.14 | 2.04 | 15.12 | 9.82 / 8.64 |
| | DeiT | 22.26 | 11.43 | 10.18 | 5.29 | 39.51 | 9.25 | 16.32 / 11.59 |
| | Swin | 20.55 | 17.89 | 8.09 | 17.7 | 13.55 | 49.1 | 21.14 / 13.12 |

Table 17: Transfer results when changing the linear classifier while the encoder remains fixed (variation of tSuc (%)). Patches are designed to fool the Swin-T model (Pytorch version, encoder and linear classifier). The transferability is measured when targeting a new network (same encoder, different linear classifier). Results are averaged over classes and over patch sizes.

| Method | Variation of tSuc (%) |
|---|---|
| GAP (Brown et al., 2017) | - 61.4 |
| LaVAN (Karmon et al., 2018) | - 42.6 |
| TTP Naseer et al. (2021) | - 51.8 |
| **Ours** $(\mathbf{SW}_2^2)_{500}^{(1)}$ | **- 0.27** |
| **Ours** $(\mathbf{W}_2^2)^{(1)}$ | **- 5.6** |

## K COMPLEMENTARY TABLES

In this section, we provide additional tables. Table 18 is the same as Table 4 present in the main paper but results are presented for different values of smoothing factors $\lambda$.

Table 18: Transfer results on robustified models by LGS defense (Naseer et al., 2019b) (tSuc (%)). Patches are designed on Swin models.

| | | Target | | | | | | mean / std |
| | | CNNs-v1 | CNNs-v2 | ENet | CNext | DeiT | Swin | |
|---|---|---|---|---|---|---|---|---|
| $\lambda = 1.5$ | Clean | 0.1 | 0.1 | 0.1 | 0.1 | 0.1 | 0.1 | 0.1 / 0 |
| | GAP (Brown et al., 2017) | 0.72 | 0.87 | 0.35 | 0.78 | 1.13 | 2.34 | 1.03 / 0.63 |
| | LaVAN (Karmon et al., 2018) | 0.56 | 0.69 | 0.3 | 0.69 | 0.82 | 2.65 | 0.95 / 0.78 |
| | L2 (Inkawhich et al., 2019) | 4.79 | 6.44 | 1.72 | 7.79 | 4.79 | 13.85 | 6.56 / 3.75 |
| | TnT (Doan et al., 2022) | 0.84 | 0.59 | 0.52 | 0.53 | 0.7 | 0.85 | 0.67 / 0.13 |
| | Casper et al. (2022) | 0.37 | 0.4 | 0.2 | 0.32 | 0.25 | 0.59 | 0.36 / 0.13 |
| | TTP (Naseer et al., 2021) | 0.68 | 0.77 | 0.28 | 0.68 | 0.76 | 1.98 | 0.86 / 0.53 |
| | M3D (Zhao et al., 2023) | 0.83 | 0.81 | 0.36 | 0.77 | 1.17 | 1.17 | 0.85 / 0.27 |
| | **Ours $(\mathbf{SW}_2^2)_{500}^{(1)}$** | **10.56** | **11.86** | **3.81** | **18.9** | **11.67** | **31.68** | **14.75 / 8.75** |
| | **Ours $(\mathbf{W}_2^2)^{(1)}$** | **13.23** | **13.4** | **4.37** | **21.42** | **13.84** | **32.08** | **16.39 / 8.58** |
| $\lambda = 1.9$ | Clean | 0.1 | 0.1 | 0.1 | 0.1 | 0.1 | 0.1 | 0.1 / 0 |
| | GAP (Brown et al., 2017) | 0.61 | 0.81 | 0.32 | 0.68 | 1. | 1.76 | 0.86 / 0.45 |
| | LaVAN (Karmon et al., 2018) | 0.45 | 0.61 | 0.26 | 0.55 | 0.72 | 1.72 | 0.72 / 0.47 |
| | L2 (Inkawhich et al., 2019) | 4.05 | 5.72 | 1.53 | 6.6 | 4.27 | 11.26 | 5.57 / 2.99 |
| | TnT (Doan et al., 2022) | 0.82 | 0.61 | 0.51 | 0.52 | 0.62 | 0.81 | 0.65 / 0.12 |
| | Casper et al. (2022) | 0.32 | 0.33 | 0.19 | 0.25 | 0.23 | 0.5 | 0.3 / 0.1 |
| | TTP (Naseer et al., 2021) | 0.56 | 0.73 | 0.24 | 0.59 | 0.66 | 1.34 | 0.69 / 0.33 |
| | M3D (Zhao et al., 2023) | 0.68 | 0.7 | 0.31 | 0.7 | 1.03 | 1.01 | 0.74 / 0.24 |
| | **Ours $(\mathbf{SW}_2^2)_{500}^{(1)}$** | **8.56** | **10.49** | **3.27** | **15.93** | **10.39** | **25.96** | **12.43 / 7.1** |
| | **Ours $(\mathbf{W}_2^2)^{(1)}$** | **10.95** | **11.98** | **3.78** | **18.37** | **12.35** | **27.07** | **14.08 / 7.19** |
| $\lambda = 2.3$ | Clean | 0.1 | 0.1 | 0.1 | 0.1 | 0.1 | 0.1 | 0.1 / 0 |
| | GAP (Brown et al., 2017) | 0.52 | 0.74 | 0.29 | 0.58 | 0.87 | 1.34 | 0.72 / 0.33 |
| | LaVAN (Karmon et al., 2018) | 0.38 | 0.55 | 0.24 | 0.47 | 0.64 | 1.19 | 0.58 / 0.3 |
| | L2 (Inkawhich et al., 2019) | 3.35 | 4.95 | 1.35 | 5.46 | 3.74 | 8.93 | 4.63 / 2.32 |
| | TnT (Doan et al., 2022) | 0.8 | 0.64 | 0.52 | 0.52 | 0.58 | 0.8 | 0.64 / 0.12 |
| | Casper et al. (2022) | 0.47 | 0.69 | 0.22 | 0.53 | 0.57 | 0.92 | 0.26 / 0.08 |
| | TTP (Naseer et al., 2021) | 0.47 | 0.69 | 0.22 | 0.53 | 0.57 | 0.92 | 0.57 / 0.21 |
| | M3D (Zhao et al., 2023) | 0.55 | 0.59 | 0.27 | 0.64 | 0.9 | 0.9 | 0.64 / 0.22 |
| | **Ours $(\mathbf{SW}_2^2)_{500}^{(1)}$** | **6.76** | **9.16** | **2.81** | **13.1** | **9.13** | **20.69** | **10.28 / 5.59** |
| | **Ours $(\mathbf{W}_2^2)^{(1)}$** | **8.85** | **10.61** | **3.27** | **15.28** | **10.86** | **22.28** | **11.86 / 5.85** |

## L  PRINTABLE PATCHES

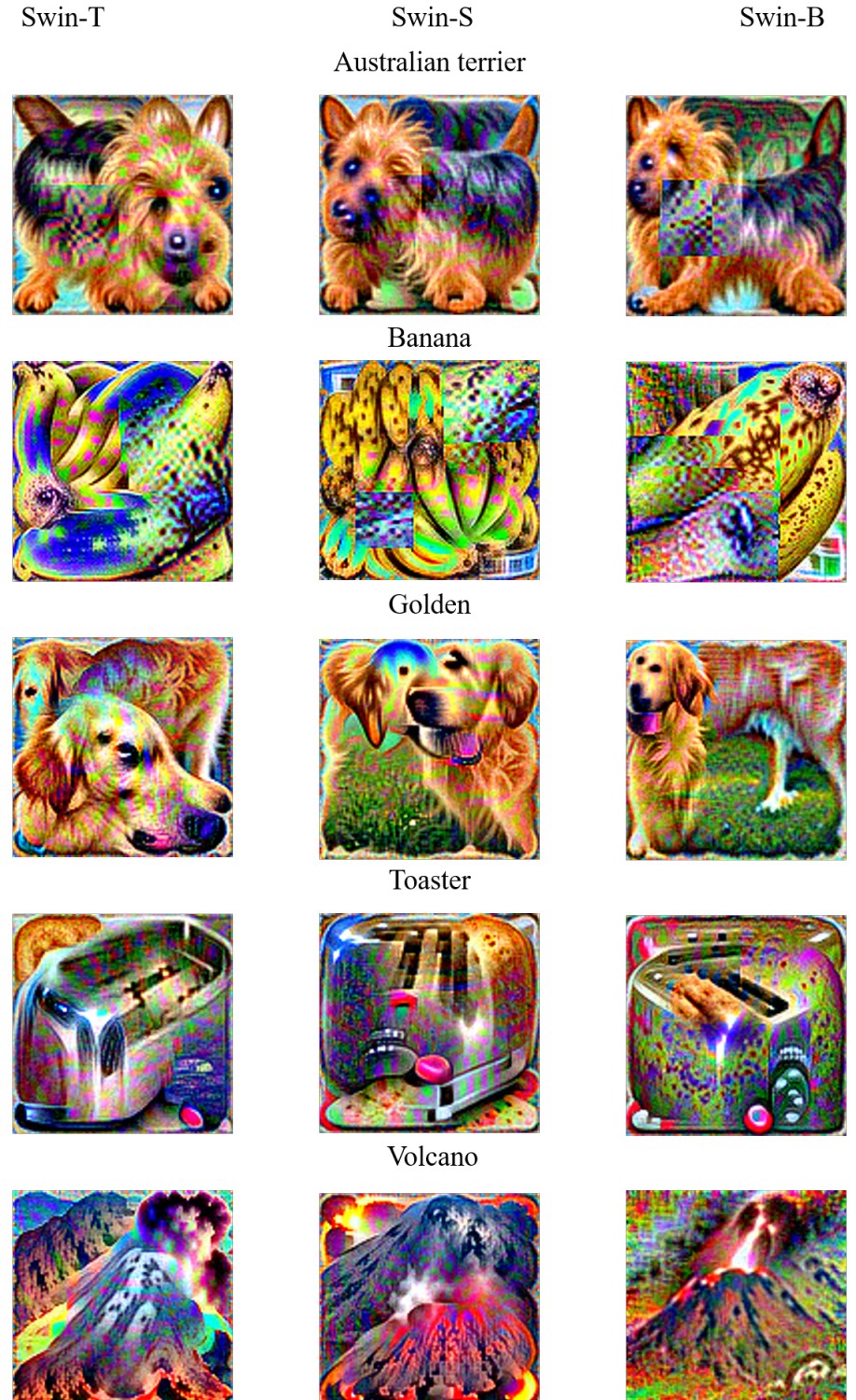

Figure 12: Printable patches designed on Swin models with our distribution-oriented method.

