# OpenReview forum: "Optimal transport based adversarial patch to leverage large scale attack transferability"
_ICLR.cc/2024/Conference — ICLR 2024 poster_

### Official Review · Reviewer_bKEx · 2023-10-27

**Soundness:** 3 good
**Presentation:** 4 excellent
**Contribution:** 3 good
**Rating:** 8
**Confidence:** 4

**Summary:**

The paper is interested in generating an adversarial patch, i.e., a small patch placed in a scene that changes the prediction of a neural network. The paper introduces an approach based on optimal transport to generate a patch, such that the estimated feature distribution of corrupted source images becomes close to the distribution of features from the target class, using either the Wasserstein and Sliced-Wasserstein distance. The black-box transferability of the attack is compared to state-of-the-art attacks for the Image Classification task on ImageNet-1K, on a broad set of networks. Qualitative results with a printed patch are also presented to demonstrate the real-world applicability of the attack.

**Strengths:**

- **Broad Model Evaluation:** The paper offers a comprehensive evaluation of attacked models, ranging from classical Convolutional Neural Networks (CNNs) to the more contemporary Vision Transformers. This large coverage is important, as it shows the adaptability and effectiveness of the proposed attack across diverse neural network architectures. Furthermore, as mentioned in the paper, recent networks and training recipes are both naturally more robust than older CNNs, it is then fundamental to evaluate against them as well. From the results, we can see that attacks optimized for a specific category of models does not transfer as efficiently to all other categories.
- **High Transferability Results:** The proposed approach demonstrates strong transferability results, outperforming in general both state-of-the-art patch and non-patch attacks. Notably, the approach accomplishes this while maintaining a similar level of computational efficiency as other existing methods.
- **Clarity and Reproducibility:** The paper is well written and easy to follow, with a clear description of the proposed approach and of the experiments. Additionally, the code is available as supplementary material which helps reproducibility.

**Weaknesses:**

- **Data Dependency:** One limitation currently not discussed in the paper is the target data requirements. The proposed attack method relies on a substantial number of target examples to generate the patch to accurately approximate the target class distribution. This means that the attack has access to a lot more information compared to attacks that matches a single feature point. This data dependency could be an important limitation in practice.

**Questions:**

I would like to see a study on the strength of the proposed attack, depending on the number of target images used to approximate the target distribution. How many target images are used to generate the patches in the experiments ? The paper mentions 40000 images to train the patches, but I assume that they are also split into source and target images.

---

> ### Author Response · Authors · 2023-11-15
> **Response to bKEx**
>
> We appreciate the feedbacks of the reviewer concerning various aspects of our work. Please find bellow our response.
>
> > Concerning the data dependency to construct the target class distribution.
>
> To generate patches, we split the validation set of ImageNet into a training set on which we train patches and into a test set on which we evaluate them. To model the target class distribution, we select images from targeted class included in training set of ImageNet dataset (composed in practice by at most 1300 images). In this regard, we agree with the reviewer that our method has more information than single feature point method, but this information is available as the model was trained on it. In Figure 9 in Appendix we show the evolution of the transferability of out attack with respect to the number of points used to model the target class distribution. As been shown, the targeted success rate (tSuc) increases with the number of considered target points. This figure is added to the supplementary material in the ablation study section. Thank you for pointing out this interesting experiment.

---

> > ### Comment · Reviewer_bKEx · 2023-11-20
> > **Answering back**
> >
> > Thanks to the authors for their answer and taking the time for the additional experiments. However, there are two comments I would like to make from the response.
> > - First, I'm not totally satisfied with the presentation of the results. As such, it is difficult to compare with other APA, since the setting is slightly different from the tables presented and the results are averaged over all networks. I would like to see where results with other APA would land on Figure 9, when compared in the same setting, to have an idea of the number of target points required to reach better results. It makes sense that the more points to estimate the distribution, the better, but what would be a good number of data points ?
> > - I do not fully agree that "this information is available as the model was trained on it". If an attacker were to create a patch to disrupt a model deployed in production, they would not have access to the training data of the model. Then, the attacker would need a thousand images of the target class to estimate the target feature distribution and create their patch using the proposed approach, whereas they would need a single image of the target class for feature-point based attacks. This remark does not invalidate the strength of the attack, but rather provides an explanation of the strong success rate observed, and it is a practical limitation not mentioned in the paper.

---

> ### Author Response · Authors · 2023-11-21
> **Thank you for the response**
>
> We thank the reviewer for the response. Please find bellow our responses.
>  -   We appreciate the feedback concerning Figure 9. To compare exactly in the same setting and to enhance the clarity of Figure 9, we add three additional runners-up baselines (GAP, LaVAN and TTP), and the supplementary material is updated correspondingly.
> Considering a larger number of targeted data points smooths the patch optimization and reduces the performance variability of the obtained patch (this result joins the results presented in Appendix B). We agree with the reviewer that a larger number of data points leads to a better estimation of the target class distribution and then to stronger patches. Figure 9 shows that considering ten target points significantly increases the patch transferability. However, the attacker may prefer to consider a larger set of target points for statistical soundness.
> -    We agree with the reviewer that the targeted model is unknown, and the dataset on which the targeted model was trained is also generally unknown. In our previous response, we were referring to the training data of the source model. We followed the common practice of transferability works where the attacker may use images from the training set of the source model to build the corresponding method (L2-based method, TTP and M3D). However, we agree with the reviewer that our method uses more information than the feature-point-based method. This limitation is added in the supplementary material in the paragraph discussing Figure 9.
> Thank you for the interesting discussion.

---

### Official Review · Reviewer_MKWR · 2023-10-30

**Soundness:** 3 good
**Presentation:** 3 good
**Contribution:** 3 good
**Rating:** 6
**Confidence:** 4

**Summary:**

This paper leverages the p-Wasserstein distance and the sliced-Wasserstein distance between the corrupted image distribution and the target distribution to generate transferable adversarial patches for attacking ViTs and CNNs. In this way, the proposed method pushes the corrupted feature distribution towards a target feature distribution. The authors attack a diverse set of victim models, including CNNs, ViTs, and adversarially trained models.

**Strengths:**

This paper is well-written.
The authors conduct extensive experiments to demonstrate the effectiveness of the proposed method.

**Weaknesses:**

The authors utilize the Wasserstein distance between the corrupted and the target distributions to optimize adversarial patches. This method is too simple, and there is no theoretical contribution in this paper. Besides, this paper should discuss why are distribution-based methods better than decision boundaries-based methods, feature point-based methods, and generation-based methods (e.g.TTP). Why is the Wasserstein distance suitable in optimizing adversarial patches compared to KL?
Additionally, the authors convert the generative methods (TTP) to iterative ones. Thus, the iterative targeted attacks should be also compared [1][2].
In summary, I think the contribution of this paper is under the acceptance threshold.


[1] Zhengyu Zhao, et al. On success and simplicity: A second look at transferable targeted attacks. Advances in Neural Information Processing Systems, 34:6115–6128, 2021.
[2] Wei, Zhipeng, et al. Enhancing the Self-Universality for Transferable Targeted Attacks. Proceedings of the IEEE/CVF Conference on Computer Vision and Pattern Recognition. 2023.

**Questions:**

How to obtain the target distribution $\mathcal{v}_y^{(l)}$.

---

> ### Author Response · Authors · 2023-11-15
> **Response to MKWR**
>
> We thank the reviewer for this comment, as it helps us to better clarify the contribution of our paper. To best of our knowledge, the proposed approach is the first attempt to use distribution based approach in the feature map and to show its superiority with respect to state-of-the-art approaches. To give more theoretical results, in Appendix C we give a proof on why our method generalizes the L2-based method. Our method may outperforms other decision boundary based methods as they tend to create a patch that overfits on the source model decision boundaries.
>
>
> > Concerning the superiority of optimal transport based loss over the Kullback-Leibler divergence.
>
> This point has been addressed in the general response (see first point in general response).
>
> > About converting generative methods into iterative universal ones.
>
> We convert generative methods into iterative universal methods since the objective of patches is to create one universal patch capable of fooling networks without knowing the scene where it may be placed. We thank the reviewer for providing new baselines to compare against. We convert [1], which is an image specific iterative method, into an iterative universal method. We add these new results in Table 2 of the main article. The proposed method in our paper, shows the better performances (obtained mean tSuc of 22.56\% and 7.55\% by our and [1] methods respectively). The approach proposed in [2] suggests a new data augmentation approach which is interesting. In our paper, our main contribution is based on proposition of a new loss function without considering any model independent strategy like data augmentation. In future works, we will consider to use this new data augmentation that may enhance our attack.
>
> | Method   |  min | mean | max
> | ----------- | ----------- |  ----------- |  ----------- |
> | Logit  Zhao et al. (2021)  | 2.22 | 7.55 | 26.55 |
> | Ours $(SW_2^2)^{(1)}_{500}$   | **8.93**        |  **22.56** |  **45.31** |
> | Ours $(W_2^2)^{(1)}$   | **8.09**       |  **21.14** |  **49.1** |
>
> > How to obtain the target distribution $\nu_y$.
>
> To obtain the target class distribution, we select target class images from the training set of ImageNet. We pass them into the encoder of the source model to obtain the targeted feature points which we use as target points. Formally the estimated target distribution is $\hat \nu_y^{(l)}  = \frac{1}{m} \sum_{j=1}^m \delta_{f^{(l)}(x_j)}$ where each $x_j$ are images of the targeted class $y$. This explanation will be added to the supplementary material.

---

> > ### Author Response · Authors · 2023-11-19
> > **Additional response to reviewer MKWR**
> >
> > We conduct an additional experiment to support that decision boundary-based methods learn a patch that tends to overfit on the source model classifier. For this purpose, we consider the transfer not between 2 different models but between 2 models sharing the same encoder but different classifiers. We select from the different methods patches trained to attack the source model Swin-T. On top of this Swin-T encoder, we train a new linear classifier from scratch on the ImageNet train set. This new linear classifier reaches the same level of clean accuracy as the previous classifier (from Pytorch) while being different. We measured the patch performance when targeting this new network (same encoder, different linear classifier). As expected, the transferability of decision boundary-based patches drops drastically (nearly by half) while our patches transferability remains almost the same. This experiment will be included in the supplementary material.
> >
> > | Method | Variation of tSuc (%) |
> > |------------------|--------- |
> > GAP (Brown et al., 2017) | - 61.4 |
> > LaVAN (Karmon et al., 2018) | - 42.6 |
> > TTP (Naseer et al. (2021)) | - 51.8 |
> > Ours $(SW_2^2)^{(1)}_{500}$ | **- 0.27** |
> > Ours $(W_2^2)^{(1)}$ | **- 5.6** |

---

> ### Author Response · Authors · 2023-11-22
>
> Dear reviewer MKWR,
>
> We would like to remind you that the discussion period ends in less than a day. We have made significant efforts to conduct additional experiments to provide you detailed responses to your reviews. Could you please check if our responses address your concerns or if you have further questions?
>
> Thanks in advance,
>
> The authors

---

### Official Review · Reviewer_dpUD · 2023-10-31

**Soundness:** 3 good
**Presentation:** 3 good
**Contribution:** 3 good
**Rating:** 6
**Confidence:** 4

**Summary:**

This paper considers the setting of a black-box transfer attack on image classification where an attacker does not know the target model. Instead of forcing corrupted image representations to cross the nearest decision boundaries or converge to a particular point, this paper proposes a distribution-oriented approach and relies on optimal transport to push the feature distribution of attacked images towards an already modeled distribution. This work shows that the proposed new distribution-oriented approach can lead to better transferable patches.

**Strengths:**

1. This work introduces a new framework based on optimal transport for creating patch attacks that are highly transferable to unknown networks. This framework is based on the idea of attacking feature distributions, which is less model-dependent than relying on decision boundaries and more robust to optimization artifacts than the feature point method.
2. This work shows that the proposed attack works for the most extensive spectrum of deep networks considered in the patch attack literature, such as various versions of Convolutional Neural Networks, Transformers, and adversarially trained models. The proposed method also shows transferability superiority through extensive experiments.

**Weaknesses:**

1. For qualitative experiments, this paper gives some results by selecting three objects present in ImageNet-1K (banana, cup, keyboard) and recording videos of them when one patch is placed or not next to the object. Yet, the performance of model in the physical world may be affected by the different angles or intensities of light. Thus, it would be more meaningful if this paper could provide the performances of the proposed approach with the change of angles or lights for the qualitative evaluation.
2. For digital experiments, the authors select from the previously defined families the following models and measure the attacking transferability when the resulting patch is used to fool the remaining models. The results show that the proposed approach can generate the stronger patch attack than its counterparts. Yet, the visual perception to human vision should also be considered. It would be more convincing if the image with the generated adversarial patch with the proposed method and its counterparts can be evaluated with the PSNR or SSIM as the evaluation metrics for the degree of recognition for human vision.
3. According to the table of computation time in the Appendix of this work, the proposed approach with sliced version SW and the normal W can achieve comparable efficiency with the previous methods. Note that the computation of optimal transport may need more time than those methods without OT, the authors may explain why the approach with OT to generate the adversarial patch can achieve the efficiency without the drop of performance.

**Questions:**

This work designs a new patch attack with optimal transport to narrow the distribution gap in the generation of adversarial patch and achieve the superior performance with empirical evidence. Yet, it would be better for this work to provide more details and explanations.

---

> ### Author Response · Authors · 2023-11-15
> **Response to dpUD**
>
> Thank you for your feedback on introducing a new framework and on large scale experiments.
>
> > Concerning providing additional experiments on the robustness of patches according to physical conditions.
>
> We have investigated the robustness of our produced patches against physical transformations. We have detailed this point in the general response (see point 2 in general response). Table 6 and Figure 7 in Appendix H show the results. Our patches transfer even in the worst-case scenario (far from the camera or when rotated), while other patches do not.
>
> > About the visual perception of patches to human vision.
>
> PSNR and SSIM metrics are commonly used to measure the potential detectability of perturbations induced by invisible adversarial examples. Adversarial Patch Attacks (APAs) rely on adding a visible textured patch to the scene: metrics such as PSNR and SSIM are not adequate in this context, because they would produce high values when comparing clean and attacked images. We want to emphasize that the purposes of invisible adversarial examples and adversarial patch attacks are different: APAs do not attempt to be made invisible by humans, as they consist of a strong image modification with a limited spatial extent. They are potentially detectable by humans as an object or region in an image but are expected to fool a deep neural-network and to be physically realizable.
>
> > Regarding the computational time of our methods.
>
> We agree with the reviewer that the exact optimal transport solved by linear programs is known to be very slow. As said in the main article, this method scales with the considered number of samples and in our case this number is low. The number of samples for source distribution ($\mu$) is $n=50$ and for the target distribution ($\nu_y$) is $m=1300$. It explains why our method does not show numerical overflow compared to other methods.
>
> > On providing more details and explanations on why our method outperforms other methods.
>
> In Appendix C, we provide a proof to show that our method generalizes the L2-based method. In the main article we try to provide some insights on why our method outperforms decision boundary based methods. Our method may outperforms other decision boundary based methods as they tend to create a patch that overfits on the source model decision boundaries.

---

> ### Author Response · Authors · 2023-11-22
>
> Dear reviewer dpUD,
>
> We would like to kindly remind you that the discussion period ends in less than a day. We have made efforts to provide new physical experiments, detailed responses to your reviews and updated our paper following your recommendations. Could you please check if our responses address your concerns or if you have further questions?
>
> Thanks in advance,
>
> The authors

---

### Official Review · Reviewer_Ctvh · 2023-10-31

**Soundness:** 3 good
**Presentation:** 3 good
**Contribution:** 3 good
**Rating:** 6
**Confidence:** 4

**Summary:**

This paper presents a distribution based approach for adversarial patch attacks. Instead of optimizing the patch to cross decision boundaries or converge to a specific point, the proposed method uses optimal transport to push the feature distribution of attacked images towards a known distribution. The paper demonstrates that this distribution-oriented approach leads to better transferable patches that can influence multiple models and can be effective in physical world experiments. The paper provides comprehensive digital, hybrid, and physical experiments to validate the effectiveness and transferability of the proposed method.

**Strengths:**

The paper introduces a novel approach for designing adversarial patch attacks based on optimal transport and distribution, which is a unique and innovative idea.

The author conducted comprehensive experiments, including in virtual, hybrid, and physical environments. The results demonstrate that the method outperforms the baseline in terms of transferability.

The paper is well-structured and provides a thorough review of APA, giving readers a clear understanding of the background and related work.

**Weaknesses:**

Limited technical contribution: While I acknowledge that optimizing patches based on optimal transport theory has some novelty, the technical contribution of this paper seems rather minimal. I believe that the method's use of Wasserstein and its variants as a loss metric is a straightforward application of existing techniques. Additionally, the usage of EOT and TV Loss is common. Thus, I'm concerned that the technical contribution of this paper might be too weak for ICLR.

The experimental results lack discussions and explanations: For instance, why is it that in the experiments, the Sliced-Wasserstein distance (SW) always underperforms compared to the Wasserstein ones (W)? What causes this? I'm concerned about whether the settings for the two proposed methods are fair. Moreover, I believe there needs to be a more comprehensive ablation study and explanation regarding the choice of target layers. The authors only investigated the impact of the last three feature layers. Why did adding J-2 significantly affect performance?

Physical experiments are somewhat weak: I believe this paper fails to demonstrate the superiority of its method in physical experiments, especially since the author claims that the method provides better transferability but only compares it to a weaker baseline (L2). Additionally, could you investigate the robustness of your method when facing physical transformations? For instance, under variations in camera angles and distances.

**Questions:**

see the weakness

---

> ### Author Response · Authors · 2023-11-15
> **Response to Ctvh**
>
> Thank you for the feedback. Glad to hear that you find our idea innovative. Here are replies to points in order.
>
> > Limited technical contribution.
>
> Using an optimal transport-based loss may be straightforward, but to the best of our knowledge and considering the literature review on transferability, the proposed method is the first to generalize feature-point-based loss. To strengthen the theoretical aspect of our contribution, we give in Appendix C the proof that the exact Wasserstein loss generalizes the L2-based method.
>
> > About the different behavior and results between the Sliced-Wasserstein distance and the exact Wasserstein distance.
>
> The Sliced-Wasserstein distance and the exact Wasserstein distance are two different distances. The first is based on one-dimensional projections and the second on cost and coupling matrices. The comparison of these two methods behavior in high dimensional spaces is an active mathematical field of research and exceeds the contribution of this paper. One of the reasons that the exact Wasserstein outperforms the Sliced-Wasserstein could be that the projections in non-informative directions bias the gradient calculated using the Sliced-Wasserstein distance. In fact, the encoder is learned to build non-correlated variables to classify images. Hence, few directions retain useful information for discrimination of the target class. This induces some bias in the gradient of Sliced-Wasserstein loss, which may cause the patch to fall into a poor local minimum consequently. However is this paper, we want to emphasis that our contribution is to show that distribution based loss has several advantages over other methods and in practice may outperforms them.
>
> > Concerning the comparison setting between the two Wasserstein distances.
>
> We try to make the comparison between the two Wasserstein distances as fair as possible. In the main article and in the supplementary material, we evaluate the performance of each method for a broad scale of different hyper-parameters. For example, in the supplementary material, we evaluate the performance of the Sliced-Wasserstein distance according to multiple number of random projections $K$ alongside its other hyper-parameters.
>
> > About the choice of the target layers.
>
> We thank the reviewer for the careful reading of the paper. We have encountered that there is a typo in the text. $l_{J-2}$ and $l_{J-1}$ correspond in fact to $l_{J-8}$ and $l_{J-2}$ respectively. These layers represent the second and the third block of Swin models, which is composed by four blocks in total (see Figure 8 in Appendix I). This experiment shows that the last layers outperforms from pushing mid layers distributions.
>
> > Only comparing against the L2 method during the physical evaluation.
>
> During our physical experiments, as said in the main article (see Section 4.3), we compare against all the baselines. However, the L2-based method and our methods are the only to show significant transferability. Hence, for the brevity, we have shown only their corresponding results. To make it more clear, we have rewritten the corresponding sentence.  We have investigated the robustness of our produced patches against physical transformations. Table 6 and Figure 7 in Appendix H show the results. Our patches transfer even in the worst-case scenario (far from the camera or when rotated), while other patches do not.

---

> ### Author Response · Authors · 2023-11-22
>
> Dear reviewer Ctvh,
>
> We would like to kindly remind you that the discussion period ends in less than a day. We have made efforts to provide detailed responses to your reviews and updated our paper following your recommendations. Could you please check if our responses address your concerns or if you have further questions?
>
> Thanks in advance,
>
> The authors

---

### Official Review · Reviewer_edp6 · 2023-11-01

**Soundness:** 2 fair
**Presentation:** 3 good
**Contribution:** 2 fair
**Rating:** 5
**Confidence:** 4

**Summary:**

The authors introduce a framework based on optimal transport for crafting patch attacks that are highly transferable to unknown networks. This framework is based on the idea of attacking feature distributions, which is claimed to be less model-dependent than relying on decision boundaries and more robust to optimization artifacts than the feature point method.

**Strengths:**

I believe the main strength of this article is that it provides an implementation based on optimal transport for distribution matching-based transfer methods.

**Weaknesses:**

I believe the main drawback of this article is that it does not discuss other methods based on distribution matching.
The article's main claim is somewhat ambiguous; is it advocating that distribution-matching methods are superior, or that methods based on optimal transport are superior? A comparison with other distribution-matching methods would make the paper more compelling.
Another issue is that the article's statements are not sufficiently rigorous. The statements seem to suggest that the method presented can relax dependency on specific models. However, in reality, the method still relies on a particular model, even though it exhibits better transfer performance in experiments.

**Questions:**

1. Compare with other distribution-matching methods, such as algorithms other than OT (Optimal Transport).
2. Revise the description concerning model dependency.

---

> ### Author Response · Authors · 2023-11-15
> **Response to edp6**
>
> We thank the reviewer for the review. Please find bellow the different clarifications.
>
> > About the comparison against different distribution matching methods.
>
> We thank the reviewer for identifying this ambiguity in the text. As this point has been also raised by other reviewers, we have provided a general response (see our general response point 1). Moreover, we want to emphasize that even if the exact and the sliced optimal transport are based on optimal transport, they have different behaviors. The sliced OT relies on one-dimensional projections and the exact OT on a cost and coupling matrix. Theses differences are noticeable in high-dimension space, like the feature space of networks. Our experiments evaluate these two different methods and show their superiority over other methods.
>
> > Revise the description concerning model dependency.
>
> We thank the reviewer for pointing out that the model dependency statement may be ambiguous. We want to add that our method is by design, independent of the classification layer. For example, our method allows the training of the patch on self-supervised networks without requiring training an extra classifier on top of it. We have clarified it in the main article (see Section 1 Introduction).

---

> ### Author Response · Authors · 2023-11-22
>
> Dear reviewer edp6,
>
> We would like to kindly remind you that the discussion period ends in less than a day. We have made efforts to provide detailed responses to your reviews and updated our paper following your recommendations. Could you please check if our responses address your concerns or if you have further questions?
>
> Thanks in advance,
>
> The authors

---

### Author Response · Authors · 2023-11-15
**General response to reviewers**

We want to thank the reviewers for their comments and suggestions. We are glad to hear that our paper is:

- providing extensive experiments showing the transferability between various families of computer vision models of our attack (dpUD, Ctvh, MKWR and bKEx).

- well written and easy to follow (Ctvh, MKWR and bKEx)

In addition to individual responses, here are some general points we would like to address:

> Both reviewers edp6 and MKWR wonder on the use and the superiority of optimal transport-based losses over the Kullback-Leibler divergence.

Our objective is to compute a meaningful criterion in the feature space that captures the geometry of the different distributions. As said at the beginning of Section 3.1, we rely on optimal transport-based losses for the following two reasons:

- OT losses (both exact and sliced version) take into account the underlying metric space (through the cost matrix) on which the probability distributions are defined,
- for non-overlapping distributions such as ours, the Kullback-Leibler divergence is infinite.

More details concerning the advantages of OT over other methods could be seen in Arjovsky et al. (2017) (Part 2: Different Distances). For the previously mentioned reasons, the KL divergence can only be used on top of the classification layer. The KL divergence is then computed on probability class distributions and not on the feature points themselves. This KL-based method was proposed by Naseer et al. (2021) (TTP), we evaluate it in the main article and it has been outperformed by our proposed method. To avoid to rely on a particular classifier and to capture the geometry of attacked and class-target images in the feature map of the source model, the natural choice is to choose OT distances as distribution matching methods. In the Appendix D, we have added a new section to discuss the advantages of using optimal transport-based losses over the KL divergence. A toy example is presented to demonstrate the different behavior between the KL divergence and the exact OT.

> Reviewers ctvh and dpUD on more physical experiments to evaluate the robustness of our patch to physical conditions.

We agree with the reviewers that a physical implementation can be affected by various factors. However, in this paper, our contribution was on the transferability aspect of patch attacks, which is a prerequisite for threatening real systems. To respond to the reviewers suggestion, we add a new Section in Appendix in which we demonstrate the robustness of our patches according to different physical conditions (see Section H in Appendix for more details and visualizations).

---

### Meta-Review · Area_Chair_wcvk · 2023-12-07

**Metareview:**

The work uses the Wasserstein distance between  corrupted image distribution and target distribution to
generate transferable adversarial patches for attacking some specific NN architectures.

Most reviewers found the paper interesting and introducing a novel and relevant methodology. Rebuttal has
addressed most of their concerns and as such, we think that the paper can be accepted to the conference.

**Justification For Why Not Higher Score:**

The paper lacks a bit of theoretical support

**Justification For Why Not Lower Score:**

The paper still proposes some novelties and is nicely executed.

---

### Decision · Program_Chairs · 2024-01-16

Accept (poster)